# Spatially defined multicellular functional units in colorectal cancer revealed from single cell and spatial transcriptomics

Inbal Avraham-Davidi[1]*[†], Simon Mages[1,2][†], Johanna Klughammer[1,2][†], Noa Moriel[3], Shinya Imada[4], Matan Hofree[1], Evan Murray[5], Jonathan Chen[5,6,7], Karin Pelka[5,6][‡, §], Arnav Mehta[5,6,8], Genevieve M Boland[5,9], Toni Delorey[1], Leah Caplan[1], Danielle Dionne[1], Robert Strasser[2], Jana Lalakova[10], Anezka Niesnerova[10], Hao Xu[10], Morgane Rouault[10], Itay Tirosh[11], Hacohen Nir[5,6], Fei Chen[5,12], Omer Yilmaz[4,13], Jatin Roper[4]*[#], Orit Rozenblatt-Rosen[1]*[¶], Mor Nitzan[3,14,15]*, Aviv Regev[1,4]*[¶]

[1]Klarman Cell Observatory, Broad Institute of MIT and Harvard, Cambridge, United States; [2]Gene Center and Department of Biochemistry, Ludwig-Maximilians-Universität München, Munich, Germany; [3]School of Computer Science and Engineering, The Hebrew University of Jerusalem, Jerusalem, Israel; [4]Departmen of Biology at MIT, Koch Institute for Integrative Cancer Research at MIT, Cambridge, United States; [5]Broad Institute of Massachusetts Institute of Technology and Harvard, Cambridge, United States; [6]Massachusetts General Hospital (MGH) Cancer Center, Harvard Medical School (HMS), Boston, United States; [7]Department of Pathology, MGH, Boston, United States; [8]Dana-Farber Cancer Institute, Boston, United States; [9]Department of Surgery, MGH, Boston, United States; [10]10xGenomics, Stockholm, Sweden; [11]Department of Molecular Cell Biology, Weizmann Institute of Science, Rehovot, Israel; [12]Harvard Stem Cell and Regenerative Biology, Cambridge, United States; [13]Department of Pathology, Massachusetts General Hospital, Boston, United States; [14]Racah Institute of Physics and Faculty of Medicine, The Hebrew University of Jerusalem, Jerusalem, Israel; [15]Faculty of Medicine, The Hebrew University of Jerusalem, Jerusalem, Israel

*For correspondence:
inbalavr@gmail.com (IA-D);
jatin.roper@duke.edu (JR);
orit.r.rosen@gmail.com (OR-R);
mor.nitzan@mail.huji.ac.il (MN);
aregev@broadinstitute.org (AR)

[†]These authors contributed equally to this work

Present address: [‡]Gladstone-UCSF Institute of Genomic Immunology, Gladstone Institutes, San Francisco, United States; [§]Department of Microbiology and Immunology, UCSF, San Francisco, United States; [#]Department of Pharmacology and Cancer Biology and Department of Medicine, Division of Gastroenterology, Duke University, Durham, United States; [¶]Genentech, 1 DNA Way, South San Francisco, United States

## eLife Assessment

This work presents a **valuable** resource combining scRNA-seq and spatial transcriptomics studies to map mouse pre-clinical models of colorectal cancer, identifying distinct cellular programs and micro-environments that could enhance patient stratification and therapeutic approaches in colorectal cancer. While the evidence provided in the manuscript are not fully validated, these **solid** data were collected and analyzed using a validated methodology that will be of interest to the community in future studies.

**Abstract** While advances in single-cell genomics have helped to chart the cellular components of tumor ecosystems, it has been more challenging to characterize their specific spatial organization and functional interactions. Here, we combine single-cell RNA-seq, spatial transcriptomics by Slide-seq, and in situ multiplex RNA analysis to create a detailed spatial map of healthy and dysplastic colon cellular ecosystems and their association with disease progression. We profiled inducible genetic CRC mouse models that recapitulate key features of human CRC, assigned cell

types and epithelial expression programs to spatial tissue locations in tumors, and computationally used them to identify the regional features spanning different cells in the same spatial niche. We find that tumors were organized in cellular neighborhoods, each with a distinct composition of cell subtypes, expression programs, and local cellular interactions. Comparing to scRNA-seq and bulk RNA-seq data from human CRC, we find that both cell composition and layout features were conserved between the species, with mouse neighborhoods correlating with malignancy and clinical outcome in human patient tumors, highlighting the relevance of our findings to human disease. Our work offers a comprehensive framework that is applicable across various tissues, tumors, and disease conditions, with tools for the extrapolation of findings from experimental mouse models to human diseases.

## Introduction

The spatial organization of diverse cells in the tumor ecosystem impacts and drives interactions between malignant cells and neighboring immune and stromal cells, either promoting or suppressing tumor growth (*McAllister and Weinberg, 2014*). Recent studies have shown that systematic understanding of the spatial organization of tumors can shed light on disease progression and response to therapy, with specific features correlated with tumor subtypes (*Pelka et al., 2021*; *Wagner et al., 2019*; *Hunter et al., 2021*), cancer prognosis (*Keren et al., 2018*; *Schürch et al., 2020*; *Jackson et al., 2020*), or response to treatment (*Jerby-Arnon et al., 2018*; *Grünwald et al., 2021*).

However, genome-scale, high-resolution dissection of the spatial organization of tumors and its functional implications remains challenging, largely due to technical limitations. Methods such as fluorescent in situ hybridization (FISH) and immunohistochemistry can only measure a handful of preselected transcripts or proteins, whereas single-cell RNA-seq (scRNA-seq) does not directly capture spatial relations. Recent advances in spatial genomics and proteomics allow multiplexed or genome-scale measurements in situ (*Goltsev et al., 2018*; *Angelo et al., 2014*; *Giesen et al., 2014*; *Ståhl et al., 2016*; *Rodriques et al., 2019*; *Stickels et al., 2021*; *Marx, 2021*; *Waylen et al., 2020*), but with a trade-off between genomic scale and spatial resolution (*Palla et al., 2022*). As a result, data from different experimental methods need to be integrated for a comprehensive view of the tissue biology. Many analytical tools have been developed to integrate some crucial aspects of the data (*Cable et al., 2022*; *Kleshchevnikov et al., 2022*; *Mages et al., 2023*; *Lopez et al., 2022*; *Biancalani et al., 2021*), but it can be challenging to deploy them and distill answers to specific disease biology questions. This leaves open many fundamental questions about tissue organization and collective function, including whether there are canonical functional units in tumors, what may be their organization in the tumor landscape, and what role each plays in tumor progression.

A case in point is colorectal cancer (CRC), where initial lesions (adenomatous polyps) progress over time to carcinoma and eventually to metastatic disease. While the mutations that drive this process were extensively studied (*The Cancer Genome Atlas Network, 2012*; *Kwong and Dove, 2009*; *Fearon, 2011*; *Fearon and Vogelstein, 1990*), and the cellular ecosystem of CRC has now been deeply charted (*Pelka et al., 2021*; *Chen et al., 2021*; *Becker et al., 2022*), the spatial landscape is less well-characterized. In a recent study of human CRC (*Pelka et al., 2021*), we statistically associated cell profiles across tumors and showed that they map to different cellular communities, reside in different locations in the tumor, and reflect different tumor subtypes. However, absent genome-wide in situ measurements, these statistical inferences do not yet reflect the full spatial organization of the tumor. Moreover, it is important to relate such patterns to those in animal models used in mechanistic studies and as pre-clinical models, to understand the relation between lab models and human tumors.

Here, we deciphered the spatial and cellular organization of CRC by combining scRNA-seq, spatial transcriptomics by Slide-seq, and in situ RNA multiplex analysis, using a novel computational framework. We first profiled two inducible genetic mouse models of CRC that recapitulate key features of human CRC, before and after tumor initiation. We integrated the spatial and cell profiles to create a spatial cell map of the tumor landscape, revealing dysplasia-specific cellular layout and potential physical interactions. We found that the tumor landscape is organized in cell neighborhoods, each with distinct epithelial, immune, and stromal cell compositions, and governed by different gene programs. Three of the cell neighborhoods are associated with tumor progression, each activating different biological pathways but all active simultaneously, albeit in different parts of the tumor. We devised

a computational framework, based on the TACCO (*Mages et al., 2023*) method, extending it to compare single-cell and spatial features of tumors between species and applied it to scRNA-seq data from human CRC. Multiple features were conserved between tumors in the mouse model and the human patients, and the mouse cellular neighborhoods correlated with malignancy and clinical outcome (progression-free intervals [PFI] and overall survival [OS]) in human patient tumors. Our work provides a general approach that can be applied to other tissues, tumors, and disease conditions.

## Results

### A cell atlas of genetic models of colorectal cancer

To chart the cell ecosystem of CRC and how it changes during tumor progression, we studied two genetic mouse models of CRC, one with inactivation of *Apc* and another in which *Apc* inactivation is accompanied by an oncogenic *Kras*^G12D/+^ mutation and inactivation of *Trp53* (*Roper et al., 2017*; *Roper et al., 2018*; *Golovko et al., 2015*; *Figure 1A*). In the AV model, Apc^fl/fl^Villin^creERT2^ mice are injected with 4-hydroxytamoxifen to the submucosal layer of the colon, inducing the deletion of Apc specifically in epithelial cells within the injection site (*Roper et al., 2017*). In the AKPV model (Apc^fl/fl^; LSL-Kras^G12D^; Trp53^fl/fl^; Rosa^26LSL-tdTomato/+^; Villin^CreERT2^ mice, Methods), 4-hydroxytamoxifen injection also induces an oncogenic Kras^G12D/+^ mutation and then inactivation of Trp53. In both cases, 4-hydroxytamoxifen injection leads to the formation of local lesions that resemble human dysplastic lesions (*Roper et al., 2017*).

We first generated a single-cell atlas of the models consisting of 48,115 high-quality scRNA-seq profiles from normal colon, AV (3 weeks after 4-hydroxytamoxifen induction), and AKPV (3 and 9 weeks after induction) tissues. We captured a diverse cell census (*Figure 1B and C*, Methods), with 35 clusters annotated post hoc by the expression of known marker genes (*Figure 1B*, *Figure 1—figure supplement 1A and B*, Methods), across epithelial, immune, and stromal cell compartments (*Figure 1—figure supplement 1C*). For validation purposes, we also generated multiplex in situ RNA profiles (with the Cartana method *Gyllborg et al., 2020*) in six sections each from normal and AV conditions, using a panel of 66–180 marker genes chosen to best represent the cell types and programs found in the tissues (*Figure 1E*, Methods).

### Tumorigenic genotypes cause shifts in the composition of epithelial cell populations and their microenvironment in AV and AKPV lesions

Dysplastic lesions exhibited shifts in proportions of immune and stromal cells, including changes in subsets pre-existing in normal tissue, as well as infiltration of new cell subsets (*Figure 1C and D*, *Figure 1—figure supplement 1D, E and F*). This resulted in both an increase in cells of existing populations (e.g. some T cell subsets [TNK05 (GdT/Il17+), TNK06 (Treg)]) and emergence of new dysplasia-associated cells (e.g. granulocytes [Gran01, Gran02] and monocytes [Mono02, Mono03]) mirroring observations in human CRC (*Pelka et al., 2021*), breast cancer (*Hagerling et al., 2019*), and non-small cell lung cancer (*Arenberg et al., 2000*; *Figure 1C and D*, *Figure 1—figure supplement 1D and E* and *Figure 1—figure supplement 2A–C*). We validated these patterns using multiplex in situ RNA analysis (*Figure 1E*, *Figure 1—figure supplement 1F–H* and *Figure 1—figure supplements 3 and 4*). Infiltration is likely to underlie many of these changes as many of the increasing cell subsets (granulocytes, monocytes, mast cells) expressed genes, such as Sell and Ccr2, indicating tissue recruitment, and as the cells dramatically increase in proportion despite negligible signals of proliferation programs.

Two of four monocyte subsets, Mono02 and Mono03, were unique dysplasia-associated cells (*Figure 1—figure supplement 2D–F*) and were respectively enriched for general inflammatory response genes (FDR = 5.7 $10^{-30}$, two-sided Fisher's exact test in GO term enrichment) and interferon beta and gamma response genes (FDR = 3.5 $10^{-11}$, 1.0 $10^{-13}$). T cell subsets showed the expected diversity across nine subsets (*Figure 1—figure supplement 2G–I*; *Smith and Garrett, 2011*), with a significant decrease (out of all T cells) in TNK01 (GdT/Cd8) in the dysplastic microenvironment and an increase in TNK05 (GdT/Il17+) (*Figure 1—figure supplement 2J*). This is consistent with the T cell composition in tumors from mismatch repair proficient (MMRp) CRC patients (*Figure 1—figure supplement 2K*). RNA velocity analysis (*La Manno et al., 2018*) of T cells from normal and AV tissue (*Figure 2A*) showed a change in inferred cellular relationships with TNK03 (naive T) and TNK02 (Th1/

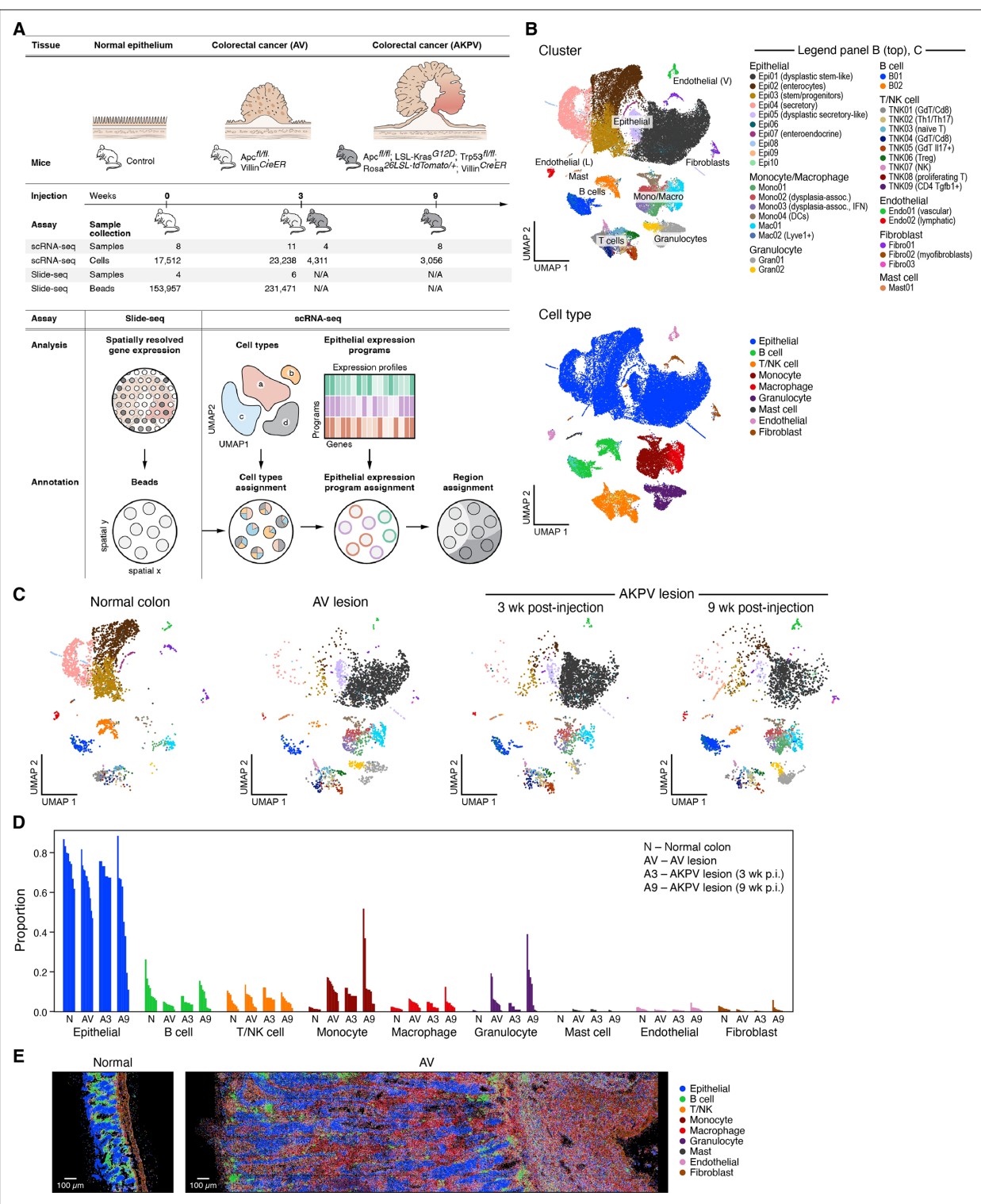

**Figure 1.** A single cell atlas of healthy colon and dysplastic lesions in mouse. (**A**) Study overview. (**B**) Major cell subsets of healthy colon and dysplastic lesions. 2D embedding of 48,115 single cell profiles colored by cluster (top, legend) or annotated cell type (bottom, legend). (**C, D**) Changes in cell composition in dysplastic tissues. (**C**) 2D embedding of single cell profiles, showing only the cells in each condition state, subsampled to equal numbers of cells per condition state, colored by cluster (same legend as in B [top]). (**D**) Proportion of cells (y axis) of each cell type in each sample (x axis). (**E**) Multiplex RNA in situ analysis. Representative images of Cartana analysis of normal colon (left) and AV lesions (right) colored by cell type assignment (same as *Figure 1—figure supplement 4A*).

The online version of this article includes the following figure supplement(s) for figure 1:

*Figure 1 continued on next page*

**Figure supplement 1.** Marker genes, cell states, and cell types in the healthy and dysplastic mouse colon atlas.

**Figure supplement 2.** Compositional and cell intrinsic changes in stromal and immune cells.

**Figure supplement 3.** Correspondence between scRNA-seq, Slide-seq, and Cartana data.

**Figure supplement 4.** Dysplasia-associated cells in the tumor microenvironment.

Th17) preceding TNK06 (Treg), consistent with the promotion of an immunosuppressive microenvironment, and TNK08 (proliferating T) also preceding TNK02 (Th1/Th17), TNK04 (GdT/Cd8), and TNK05 (GdT/Il17+) populations.

Within the five subsets of stromal cells (including vascular endothelial and lymphatic endothelial cells and three fibroblast subsets, *Figure 1—figure supplement 2L–N* and Methods), Endo01 (vascular) were enriched in dysplastic lesions compared to normal colon (FDR = 1.8 $10^{-3}$, two-sided Welch's t test on CLR transformed compositions; *Figure 1—figure supplement 2O*). Angiogenesis-related pathways, such as angiogenesis (FDR = 1.0 $10^{-11}$) and positive regulation of angiogenesis (FDR = 3.4 $10^{-4}$) as well as glycolytic process (FDR = 1.4 $10^{-4}$) were enriched in Endo01 (vascular) from lesions compared to normal colon (*Supplementary file 1*). This is in line with vascular adaptation to the tumor's growing needs for nutrients and oxygen (*Ziyad and Iruela-Arispe, 2011*) and with the increased expression of the vascular growth factor *VegfA* in both monocytes and macrophages (*Figure 1—figure supplement 2P*).

## Cell-intrinsic expression shifts in different sub-lineages in the dysplastic epithelium

Epithelial cells showed dramatic cell-intrinsic changes between normal tissues and either AV or AKPV lesions such that the cell profiles of dysplastic epithelial cells in both models were highly distinct from normal epithelial cells (and similar to each other; *Figure 1C* and Methods). Epithelial cell profiles from normal mice (41% of cells) separated from most of those from AV and AKPV models (59% of cells; *Figure 3A*, *Figure 3—figure supplement 1A and B*), suggesting a common shift in all dysplastic cells from the normal state. Notably, 11% of epithelial cells from AV/AKPV mice were classified as non-dysplastic healthy cells, indicating that normal, non-dysplastic cells may be present in or adjacent to the lesion microenvironment (*Figure 3B*, *Figure 3—figure supplement 1A*), although we cannot firmly rule out adjacent tissue contamination. We annotated two cell clusters – Epi01 (dysplastic stem-like) and Epi05 (dysplastic secretory-like) – as dysplastic, due to their virtually exclusive presence in AV and AKPV models and because they expressed high levels of *Apc* target genes (e.g. *Axin2, Ascl2, Myc, Ccnd1, Lgr5*) and were enriched in tdTomato+ cells from AKPVT mice (*Figure 3C*, *Figure 3—figure supplement 1C and D*) while also spanning the enterocyte to secretory continuum with healthy cells (*Figure 3—figure supplement 1B*, PC2).

Interestingly, Epi05 (dysplastic secretory-like) had distinguishing markers (e.g. *Ccl9, Mmp7, Ifitm3*) from their counterparts in normal tissue, Epi04 (secretory; *Figure 3A-C*, *Figure 3—figure supplement 1A, C and E*). *Mmp7* and *Ifitm3* are known to promote metastasis in human CRC (*Zeng et al., 2002*; *Li et al., 2011*), and *Ccl9* expression by epithelial cells promotes tumor invasion through recruitment of Ccr1+ myeloid cells to the tumor's invasive front in a mouse model of CRC (*Kitamura et al., 2007*). Notably, *Ccr1* is expressed by newly recruited monocytes, macrophages, and granulocytes in our model, suggesting a potential mechanism for tumor infiltration and invasion (*Figure 3—figure supplement 1E*). Thus, dysplastic secretory epithelial cells may perform additional functions that differ from those of their healthy counterparts.

RNA velocity (*La Manno et al., 2018*) analysis of the epithelial cell compartment predicted that in the dysplastic epithelium (*Figure 2B*, right) a proliferating stem-cell-like dysplastic subset is a direct source to both a massively expanded and heterogeneous non-proliferating population of dysplastic stem-like cells (expressing WNT signaling and angiogenesis programs) and to dysplastic cells of different 'differentiation states' (MHCII expressing stem/progenitors leading to enterocytes and secretory-like cells; *Figure 2B and C*). Conversely, in the normal epithelium (*Figure 2B*, left), the 'root' is placed in a much smaller population of proliferating intestinal stem cells, and the stem cell compartment is overall much more modest. The proliferation and differentiation path of previously identified normal epithelial cells remains intact (i.e. reminiscent of the one in normal samples) even

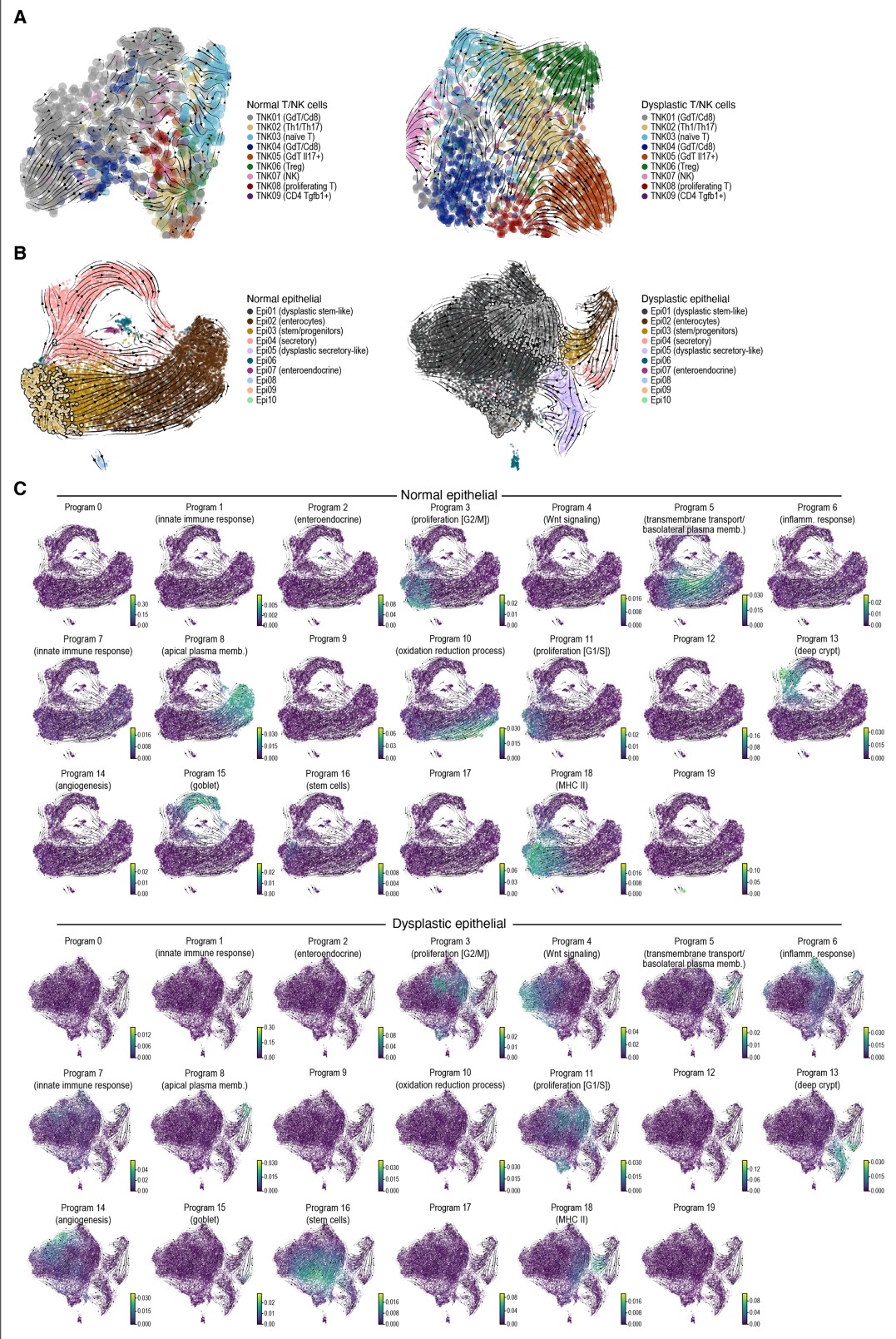

**Figure 2.** Changes in T/NK and epithelial cell differentiation in dysplastic lesions. (**A**) T/NK cell differentiation. 2D embedding of T/NK cell profiles from normal tissues (left) and dysplastic lesions (right), colored by cell subset. Streamlines: averaged and projected RNA velocities. (**B, C**) Proliferating stem-like cells give rise to an expanded stem-like compartment and differentiated-like tumor cells in dysplastic lesions. 2D embedding of epithelial cell

*Figure 2 continued on next page*

*Figure 2 continued*

profiles from normal tissues (**B**, left and **C**, top) and dysplastic lesions (**B**, right, and **C**, bottom), colored by cell subset (**B**), or by expression of epithelial programs (**C**). Streamlines: averaged and projected RNA velocities. Outlined dots (**B**): Proliferative cells (cells with more than 50% program weight in the proliferation programs #3 and #11).

in the dysplastic lesions (*Figure 2B*). While our RNA velocity analysis provides insights into potential cellular trajectories, further experimental validation is required to confirm these findings.

## Expression programs for stem-like functions, Wnt signaling, angiogenesis, and inflammation are activated in dysplastic epithelial cells

Both normal and dysplastic epithelial cells varied along a continuum, as expected and previously observed in the ongoing differentiation in the colon epithelium (*Pelka et al., 2021*; *Haber et al., 2017*; *Biton et al., 2018*; *Smillie et al., 2019*) and our RNA Velocity analysis (*Figure 2*). Using non-negative matrix factorization (iNMF from LIGER *Welch et al., 2019*, Methods), we recovered 20 expression programs spanning the different epithelial functions, and annotated them by Gene Ontology terms enriched in their top 100 weighted genes (*Figure 2C*, *Figure 3D and E*, *Figure 3—figure supplement 1F–L*, Methods).

The programs enriched in different dysplastic cells highlighted key processes that play a role in tumor promotion, including stem cell programs, Wnt signaling, angiogenesis, and inflammation and innate immunity, including interferon alpha, beta, and gamma pathways (*Figures 2C and 3D and E*). In particular, the stem cell program (#16) detected in some cells across all conditions was enriched in dysplastic samples (FDR = 5.6 $10^{-10}$, two-sided Welch's t-test on CLR transformed compositions), reminiscent of a recently described population in human (*Chen et al., 2021*; *Becker et al., 2022*). Comparing cells from dysplastic and normal samples that express the stem cell program, the dysplastic cells had distinct expression profiles with induction of negative regulators of Wnt signaling (FDR = 4.5 $10^{-5}$, two-sided Fisher's exact test in GO term enrichment, e.g. *Notum*, Wnt inhibitory factor 1 [*Wif1*] and *Nkd1*) and genes that are related to cellular response to interferon-gamma (FDR = 1.7 $10^{-6}$, e.g. *Ccl9, Ccl6*) and immune system process (FDR = 6.4 $10^{-4}$, e.g. Ifitm1 and Ifitm3; *Figure 3E and F*). This is consistent with recent studies showing that *Apc*-mutant stem cells secrete negative regulators of Wnt signaling to induce the differentiation of the WT stem cells in their proximity, thereby outcompeting them and promoting tumor formation (*van Neerven et al., 2021*; *Flanagan et al., 2021*). Thus, stem cells from dysplastic lesions may have non-canonical function and regulation. In addition, the programs for Wnt signaling (expressing both positive and negative regulators; #4, FDR = 2.8 $10^{-6}$), angiogenesis (#14, FDR = 1.2 $10^{-9}$), inflammatory response (#6, FDR = 1.4 $10^{-6}$), and innate immune response and interferon response (#7, FDR = 1.2 $10^{-2}$) were all predominantly expressed or enriched in AV/AKPV epithelium (all with two-sided Welch's t-test on CLR transformed compositions, *Figure 3E*, *Figure 3—figure supplement 1I–M*). These results are consistent with the known role of the Wnt signaling pathway in CRC, and of angiogenesis, response to hypoxia, and inflammation in tumor progression (*Clevers, 2006*; *Folkman, 2002*; *Lasry et al., 2016*).

## Malignant-like tissue programs and composition are conserved between mouse and human tumors

To evaluate the relevance of our findings to human CRC, we compared them to a scRNA-seq atlas we recently generated from tumor and adjacent normal tissue from 62 patients with either MMRp or MMRd CRC (*Pelka et al., 2021*). We compared mouse and human tumors in terms of their epithelial expression programs, cellular composition, and cell associations in multicellular hubs (*Pelka et al., 2021*). While our mouse model should more closely resemble MMRp tumors, we compared both classes separately and together to identify any shared features.

To assess the similarity between mouse and human programs, we controlled for overall cross-species and batch differences by normalizing program-specific expression profiles with species-specific background profiles (Methods). The Pearson correlation coefficients of these normalized scores between the human and mouse programs indicate some overlap in the programs defined on mouse and human

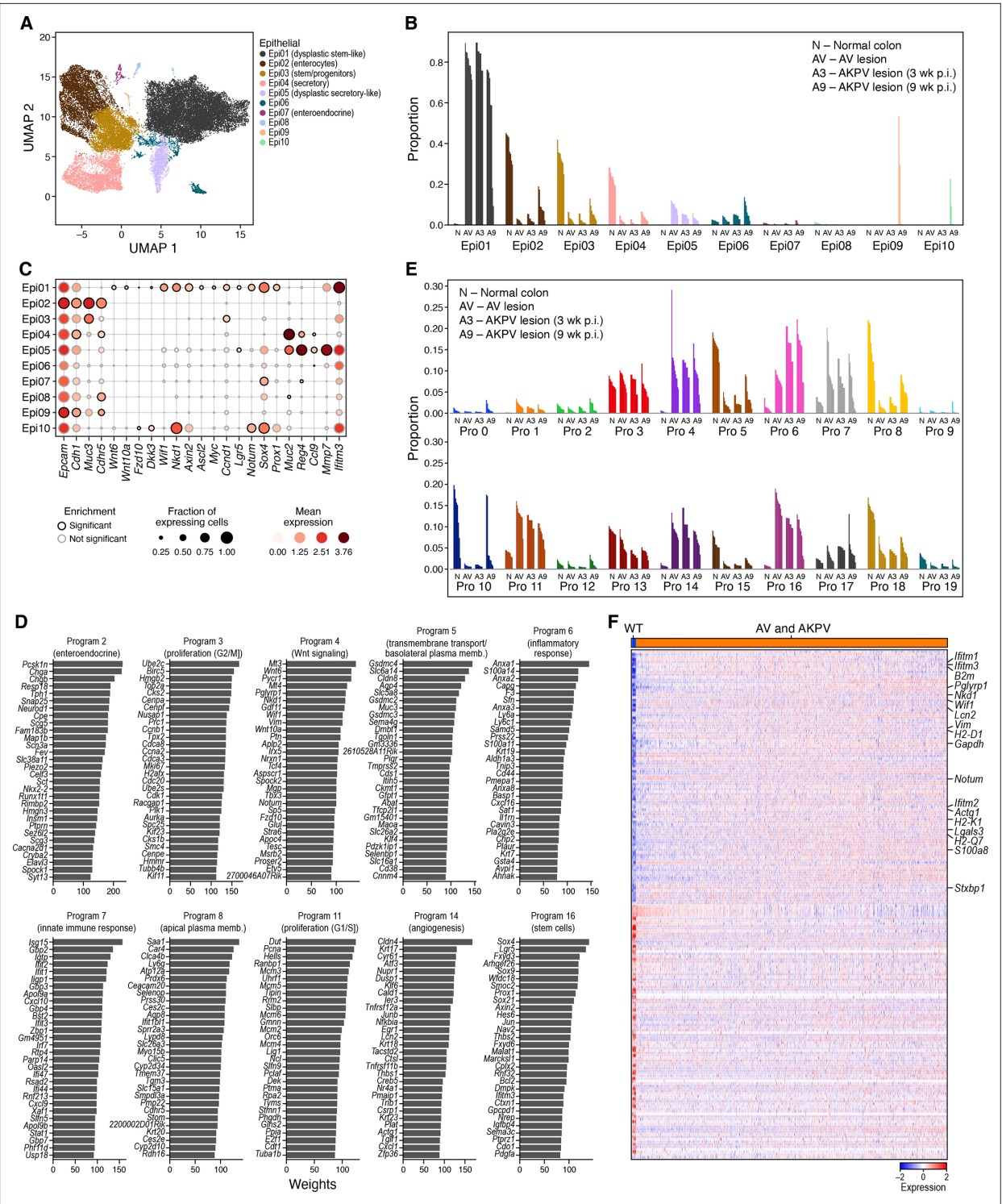

**Figure 3.** Composition and cell intrinsic expression program changes in dysplastic epithelial cells. (**A–C**) Compositional changes in epithelial cells in dysplastic tissue. (**A**) 2D embedding of epithelial cell profiles colored by clusters (legend). Cluster Epi06: potential doublets (Methods). (**B**) Proportion of cells out of all epithelial cells (y axis) of each epithelial cell subset in each sample (x axis). (**C**) Fraction of expressing cells (dot size) and mean expression in expressing cells (dot color) of marker genes (columns) for each cluster (rows). (**D–E**) Use of epithelial cell programs changes in dysplastic tissue. (**D**) Weights (x axis) of each of the 20 top ranked genes (y axis) for each program. (**E**) Proportion of program weights summed over all epithelial cells (y axis) in each sample (x axis). (**F**) Stem cell program 16 is induced in epithelial cells in dysplastic tissue. Scaled log-normalized expression (color bar) of the top 100 genes differentially expressed between cells from normal colon and from dysplastic (AV and AKPV) across the 10,812 cells that accounted for 90% of program 16's expression across all epithelial cells (columns). Selected program genes are marked.

*Figure 3 continued on next page*

*Figure 3 continued*

The online version of this article includes the following figure supplement(s) for figure 3:

**Figure supplement 1.** Changes in cell composition and expression programs usage in dysplastic epithelium.

**Figure supplement 2.** Conservation of cellular composition and expression programs between mouse and human scRNA-seq data.

data (*Figure 3—figure supplement 2A*). Epithelial cells from human and mouse tumors expressed many programs highly correlated between the species (*Figure 3—figure supplement 2B*, Methods), including for cell cycle, inflammation, epithelial secretory, angiogenesis, Wnt signaling, and normal colon functions.

Co-variation in cell proportions across samples (by scRNA-seq) was also conserved between human and mouse tumors, suggesting broad conservation of tumor composition. For example, in both species, the proportion of endothelial cells and fibroblasts correlated across samples, as did T and B cell proportions in human tumors and mouse dysplastic lesions (*Figure 3—figure supplement 2C*). Moreover, when we transferred epithelial program annotations from mouse to human scRNA-seq and calculated their co-variation across samples in each species, programs 11 (proliferation), 14 (angiogenesis), and 16 (stem cells) co-varied both across dysplastic mouse tumors and across human MMRp and MMRd tumors (*Figure 3—figure supplement 2D* and Methods), suggesting a conserved dysplastic tissue architecture.

## Integrated spatial and single-cell atlas of mouse CRC tumors

To comprehensively decipher the distribution of cells and programs in the tumor spatial niche, we next used Slide-seqV2 (*Stickels et al., 2021*) for genome-wide spatial RNA-seq at 10 µm resolution. We sectioned and profiled frozen tissues from four normal colon and four AV lesions using 10 Slide-seqV2 pucks (Methods), recovering 221,936 high-quality beads (*Figure 4A*, *Figure 4—figure supplement 1A–C*, Methods). We then integrated the single-cell census and spatial profiles using TACCO, which allowed us to annotate each bead with compositions of discrete cell types (from epithelial, immune, and stromal compartments) and to further annotate the epithelial fraction of each bead with a composition of epithelial program activity (*Figure 1A* 'annotation').

We first used TACCO to annotate every bead in the Slide-seq data with a composition of discrete cell subtypes for every puck separately, using its matching single-cell reference (normal or disease; *Figure 1A*, Methods). To this end, TACCO iteratively solved optimal transport problems to assign cell subtypes to fractions of reads of the beads. TACCO relies on unbalanced optimal transport to allow for shifts in the frequency of cell subtypes in the pucks vs. the single-cell dataset, while using the reference cellular frequencies as prior knowledge (*Figure 4—figure supplement 1E and G*). TACCO's cell type mapping recapitulated the muscularis layer in its expected tissue location based on the inferred cellular composition pattern (*Figures 4A and 5A* and *Figure 4—figure supplement 1D*). This illustrates that TACCO's compositional annotations align well with biological patterns.

Next, we used TACCO to map the epithelial gene programs (defined above), focusing on transcript counts that are inferred as derived from epithelial cells. TACCO partitioned the read count matrices for each puck, assigning counts to epithelial cells based on the mapped per-bead cell subtype annotations (from the first step) and the expression profiles associated with each subtype (Methods). It then summed all epithelial contributions into an epithelial-only spatial count matrix, followed by optimal transport to assign epithelial program contributions to individual beads, based on epithelial cell-only read signals. As for cell type mapping, the proportional contribution of the programs largely recapitulated their contributions in scRNA-seq (*Figure 4—figure supplement 1F and H*).

## Altered and less ordered local cellular organization of dysplastic lesions

We assessed the local cellular architecture in terms of the preferential proximity of cells of certain type or expressing particular epithelial programs, within a fixed-sized neighborhood, by adapting an earlier method. We defined a z-score as significance of the observed neighborhood relations compared to the null for neighborhoods of 20, 40, or 60 µm diameter (*Figure 4B and C*, *Figure 4—figure supplement 2A and B*). This z-score is defined with respect to a population of random cell type annotations generated by random permutations of the cell type annotations between the beads, where in our case we permute fractional cell type contributions.

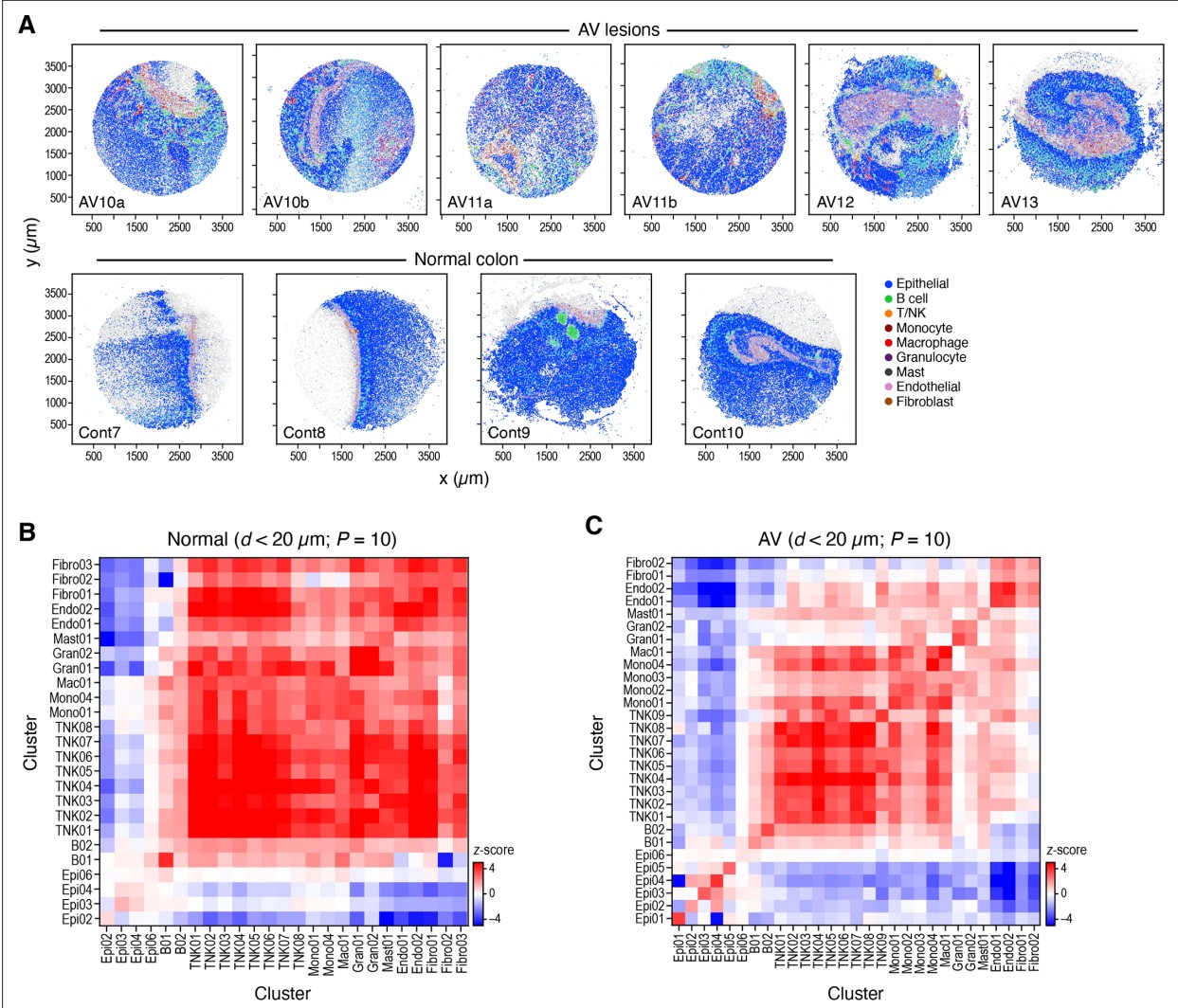

**Figure 4.** Altered cell type neighborhood in CRC. (**A**) Cell type distributions in situ. Slide-seq pucks of dysplastic (top) and normal (bottom) tissue colored by TACCO assignment of cell labels (legend, light gray: low-quality beads; x and y axis: spatial coordinates in μm). (**B, C**) Cell type neighborhoods in normal and dysplastic colon tissue. Short-range (up to 20 μm) neighborhood enrichment (Z score, color bar) vs. a background of spatially random annotation assignments for each pair of cell annotations (rows, columns) in normal (**B**) and dysplastic (**C**) tissue.

The online version of this article includes the following figure supplement(s) for figure 4:

**Figure supplement 1.** Spatial distributions of cells and programs across regions.

**Figure supplement 2.** Distinct cellular layout in normal and AV tissues.

Cell proximity preferences in the normal colon tissue are consistent with the expected morphology, validating our approach (*Figure 4A and B*). Epithelial cells were organized such that the differentiated Epi02 (Enterocytes) are excluded from the stem cell niche (*Figure 4B*, *Figure 4—figure supplement 2C*), and endothelial cells and fibroblasts were also spatially co-located in a focused region (*Figure 4B*), with T cells in their vicinity (*Figure 4B*). Our multiplex in situ RNA analysis validated the exclusion of enterocytes from the stem cell niche, as also seen in Slide-seq data (*Figure 4—figure supplement 2C and D*).

While some normal tissue features are preserved in dysplastic samples, including co-location of cells of the same lineage (*Keren et al., 2018*; *Goltsev et al., 2018*; *Figure 4A–C*), there were notable changes and more disorder. Cell types were more randomly distributed in AV lesion *vs.* normal tissue, reflected in lower z-scores (p=1.6 10$^{-37}$, one-sided Mann-Whitney U test; *Figure 4—figure supplement 2E*). At short distances, all epithelial cells (normal and dysplastic) were preferentially located close to cells from the same subtype (*Figure 4C*) and even to cells with similar functions: epithelial

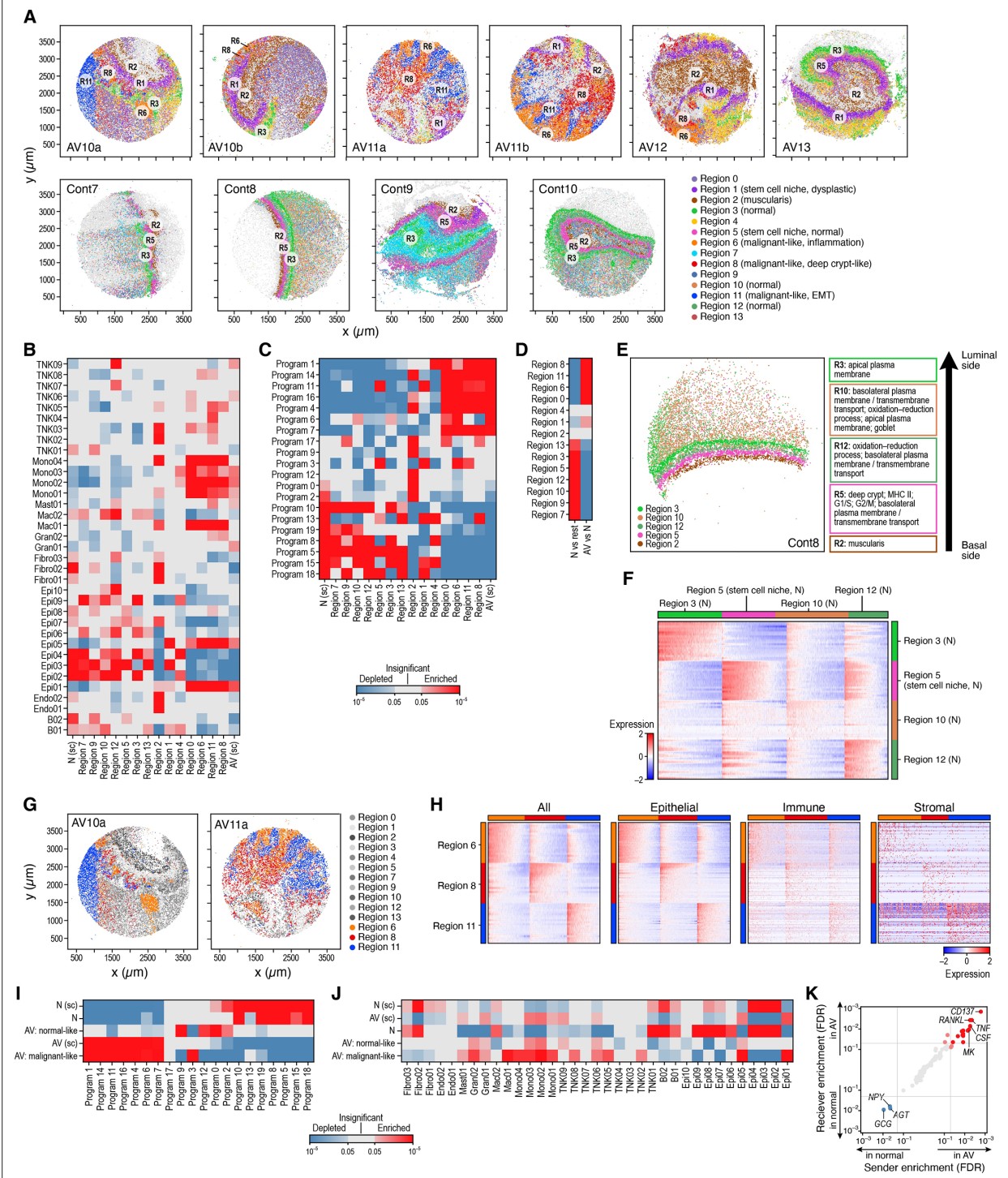

**Figure 5.** Three cellular neighborhoods associated with tumor progression. (**A**) Spatial regions. Slide-seq pucks of AV (top) and normal (bottom) mouse colon colored by TACCO regions (legend, light gray: low-quality beads; x and y axis are spatial coordinates in μm). (**B, C**) Enrichment and depletion of cell subsets and epithelial programs across different regions. Significance (FDR, color bar, two-sided Welch's t-test on CLR-transformed compositions) of enrichment (red) or depletion (blue) of specific cell subsets (rows, **B**) or epithelial cell programs (rows, **C**) in the different regions defined by TACCO (columns) as well as all normal ('N (sc)', leftmost column) and AV ('AV (sc)', rightmost column) samples from the scRNA-seq data. (**D**) TACCO defined regions preferentially relate to normal or AV tissue. Significance (FDR, color bar, two-sided Welch's t-test on CLR-transformed compositions) of enrichment (red) or depletion (blue) of each TACCO defined region (rows) in normal ('N vs. rest') and AV ('AV vs. N') samples (columns). (**E**) TACCO reveals normal colon architecture. Left: Slide-seq puck of normal mouse colon colored by TACCO region annotations (legend) (x and y axis: spatial coordinates (μm)). Right: Main epithelial expression programs enriched in each region (FDR <6.3 10⁻⁴, two-sided Welch's t test on CLR-transformed

*Figure 5 continued*

compositions) except region 2 (muscularis), which is characterized by non-epithelial (stromal) cell types. (**F**) Expression signatures of cells in normal regions 3, 5, 10, and 12. Scaled log-normalized expression of the top 20 differentially expressed genes (rows) for each bead (columns) in the region. (**G, H**) Malignant-like regions. (**G**) Slide-seq pucks of two AV lesions colored by TACCO annotations of malignant-like regions 6, 8, and 11. (**H**) Scaled log-normalized expression of the top 20 differentially expressed genes (rows) of each bead (left, columns) in the region, or of each epithelial (middle left), immune (middle right), or stromal (right) fraction of beads (columns) in regions 6, 8, and 11 in dysplastic lesions. (**I, J**) Epithelial cell subsets and programs associated with 'malignant-like', 'normal-like', and normal tissues. Significance (FDR, color bar, two-sided Welch's t-test on CLR-transformed compositions) of enrichment (red) or depletion (blue) of epithelial cell programs (**I**, columns) or epithelial, immune, and stromal cell subsets (**J**, columns) in different tissue types (rows) based on Slide-seq or scRNA-seq ('sc') samples. (**K**) Inferred interaction pathways. Enrichment (FDR) in AV vs. normal tissue of corresponding 'sender' (x axis) and 'receiver' (y axis) (aggregated over ligand-receptor pairs) pathways (dots). Red/blue: pathways significantly enriched in AV/normal samples and (light red: only for either sender or receiver). The top 5 enriched pathways (in each direction) are labeled.

The online version of this article includes the following figure supplement(s) for figure 5:

**Figure supplement 1.** Cellular neighborhoods.

**Figure supplement 2.** TLS-like structures in AV lesions by Cartana multiplex in situ RNA profiles.

**Figure supplement 3.** Ligand-receptor analysis.

cells expressing programs associated with malignant-like function (e.g. program 4 [Wnt signaling], 14 [angiogenesis], and 16 [stem cells]) resided close to each other and were spatially distant from cells expressing programs that are related to normal epithelial functions (e.g. program 5 [basolateral plasma membrane], 8 [apical plasma membrane], and 10 [oxidation-reduction process]), supporting a model where tumor progression is structured and compartmentalized (*Figure 4—figure supplement 2F*). Immune and stromal cells were generally excluded from epithelial cell neighborhoods. Granulocytes aggregated together (self-proximal; *Figure 4C*) and were relatively close to endothelial cells and dysplasia-associated monocytes (Mono02, Mono03), consistent with their recruitment from the blood through the vessels (*Figure 4C*, *Figure 4—figure supplement 2G*). Our multiplex in situ RNA analysis validated the spatial enrichment of monocytes and granulocytes near vessels (*Figure 4—figure supplement 2H*).

## Epithelial regional analysis recovers canonical structures in normal colon

To detect distinctive tissue regions in tumors, which lack traditional tissue references, we identified cellular neighborhoods with both similar epithelial program activity and a particular composition of immune and stromal cells. Specifically, we first identified 'epithelial program regions' as areas of distinct epithelial program activity and then found immune or stromal cells associated with each region (*Figure 1A* 'annotation'). Intuitively, we defined 'regions' based on both the similarity in epithelial expression program activity and proximity in space. To do this, after assigning epithelial programs to epithelial beads, we clustered the beads based on a weighted sum of spatial proximity and expression program similarity. This results in spatially contiguous annotation of beads with distinct epithelial program activity, which, together with the immune and stromal cells in their proximity, compose the 'region'. Specifically, using TACCO, we defined epithelial program regions by Leiden clustering of the weighted sum of neighborhood graphs for spatial bead proximity and epithelial expression program similarity, such that transcriptionally similar epithelial beads on different pucks can be connected (despite 'infinite' spatial distance, Methods). We then used this single framework for region annotation across all pucks (*Figure 5A*) to determine the distinctive composition of additional cell types in the same set of spatial regions (*Figure 5B-D*, *Figure 5—figure supplement 1A*).

In the normal colon, the regional analysis (*Figure 5A*, bottom) robustly recovered the expected spatial organization of the healthy colon across five regions and their cellular composition and sublayers (*Figure 1A*), from luminal/apical to basal. Four regions recovered by TACCO corresponded to different layers of the mucosa (*Figure 5A, E and F*): a luminal layer with reads found beyond the cellular layer and likely representing cellular debris trapped in the mucus; three apical layers expressing programs related to normal epithelial function (transmembrane transport, oxidation-reduction process) with gradual transition from apical to basal features; and a basal-most layer, enriched for the deep crypt, proliferation (G1/S, G2/M), MHCII and basolateral plasma membrane programs, all common features of the deep crypt area. Finally, region 2, enriched with fibroblasts, myofibroblasts, and endothelial cells, and located in the most basal side of the tissue, captured the submucosal and muscularis propria

layers, which are predominantly comprised of fibroblasts and muscle, respectively, alongside blood and lymphatic vessels, nerves, and immune cells. Overall, TACCO recovered the known organization of the colon, showing the power of our unsupervised mapping approach and shedding light on expression programs that are required for the maintenance of normal colon homeostasis.

## Dysplastic lesions maintain some of the programs of the corresponding regions in healthy tissue

AV lesions did not maintain the robust organization of normal tissues and reflected the expected histopathology of high-grade dysplasia, when dysplastic cells are confined to the mucosal layer and do not invade the submucosa (*Fleming et al., 2012*; *Figures 1A and 5A*, top). Specifically, the submucosal and muscularis propria layers from both normal and AV lesions were assigned to region 2 (*Figure 5A*).

Despite the altered morphology, some of the disrupted regions also expressed programs characteristic of their normal healthy function, suggesting that tumor progression is spatially structured and compartmentalized. For example, the region above the submucosa, captured as region 1 in AV lesions and region 5 in normal colon (*Figure 5A*), had similar features in both AV lesions and normal samples. Thus, although the overall spatial organization was disrupted in the lesion, region 1 in AV lesions expressed programs that are reminiscent of the normal deep crypt region 5 and was enriched for deep crypt cells and programs that are related to proliferation and MHC II (*Biton et al., 2018*; *Figure 5A and C*). These included proliferation programs 3 and 11 and both normal stem cells (Epi03) and dysplastic secretory-like cells (Epi05), as well as dysplastic stem cells (Epi01, although to a lesser extent than some other regions), so it may reflect one of the proliferative stem cell (and dysplastic secretory-like) niches in AV models (*Figure 5B and C*). Other regions in the AV lesions also contained some epithelial cells with normal profiles, expressing programs that should allow them to maintain their capacity to perform normal tasks. For example, region 3 expressed apical plasma membrane functions, and region 10 was enriched with oxidation-reduction functions (*Figure 5C*).

To learn about the spatial distribution of the dysplastic regions, we measured their distance from region 2 (muscularis), which is a stable landmark in the lesions. Remnants of the layered structure of the healthy tissue were still observed in the AV tissue, especially at relatively low distances from the muscularis. For example, healthy region 5 – characteristically located at distances of about 150–200 µm from the muscularis – is replaced by dysplastic region 1, peaking at 200 µm. All malignant-like regions (6/8/11) were spatially associated at ~300–700 µm from the muscularis (*Figure 5—figure supplement 1C*), located ~100–400 µm apart from each other (*Figure 5—figure supplement 1D*). We further validated this result at the protein level, by staining for β-catenin, showing an (inactive) cytoplasmic localization in the region adjacent to the muscularis, and mostly nuclear (active) localization in distal regions, near the lumen (*Figure 5—figure supplement 1B*).

## Three spatially and functionally distinct tumor regions enriched in AV lesions

Three regions – 6, 8, and 11 – had epithelial composition and programs that suggested advanced malignant-like characteristics, each highlighting a potentially different mechanism for tumor progression (*Figure 5G and H*). These three 'malignant-like regions' were enriched (vs. all other regions) in stem cell, Wnt signaling, and angiogenesis programs (16, #4, and #14; FDR = 9.6 $10^{-21}$, 8.1 $10^{-10}$, 3.7 $10^{-10}$, two-sided Welch's t-test on CLR transformed compositions) and depleted of normal epithelial programs (#5, #8, and #10; FDR = 5.5 $10^{-12}$, 1.6 $10^{-12}$, 4.2 $10^{-10}$, *Figure 5I*). Furthermore, the malignant-like regions were enriched in immune cells, including monocytes-macrophages (FDR ≤ 1.5 $10^{-5}$; excluding Mac02 [Lyve1+]), T cell subsets TNK02 (Th1/Th17), TNK05 (GdT/Il17+), TNK06 (Treg), TNK08 (proliferating T) (FDR = 2.4 $10^{-3}$, 2.2 $10^{-4}$, 1.7 $10^{-3}$, 9.6 $10^{-4}$; two-sided Welch's t-test on CLR transformed compositions), infiltrating granulocytes (FDR ≤ 9.7 $10^{-3}$), and mast cells (FDR = 1.4 $10^{-2}$), suggesting an ongoing immune response (*Figure 5J*). However, each one of the three regions had a different epithelial program composition, suggesting that in each type of region, there is a different dominant pathway/feature that may drive tumor progression (*Figure 5C*, *Supplementary file 2*).

Region 6 was characterized by an inflammatory and angiogenic multicellular community, with epithelial and immune cells expressing inflammatory programs, endothelial cells and monocytes connected in a pro-angiogenic circuit, and pro-invasive genes expressed by both endothelial and immune cells (*Figure 5B and C*). Specifically, region 6 was distinctly enriched for proliferation (programs 3 and 11;

FDR = 2.2 10$^{-14}$, 1.2 10$^{-15}$, two-sided Welch's t-test on CLR transformed compositions) and inflammatory epithelial programs (programs 6 and 7; FDR = 9.0 10$^{-7}$, 2.0 10$^{-11}$), and its non-epithelial compartment was correspondingly enriched for genes from inflammatory pathways, including the response to TNF, IL-1, and IFNγ (FDR = 3.1 10$^{-4}$, 2.8 10$^{-3}$, 4.9 10$^{-6}$, two-sided Fisher's exact test in GO term enrichment), and chemotaxis of monocytes, neutrophils, and lymphocytes (FDR = 1.5 10$^{-3}$, 3.2 10$^{-10}$, 1.0 10$^{-2}$, *Supplementary file 2*), suggesting recruitment of inflammatory cells from the circulation or other parts of the tissue. Region 6 was also enriched for collagen binding genes and collagen-containing extracellular matrix (ECM) genes (FDR = 1.4 10$^{-2}$, 7.6 10$^{-5}$, two-sided Fisher's exact test in GO term enrichment, *Supplementary file 2*), which are important for migration and invasiveness (*Winkler et al., 2020*). These include *Sparc*, expressed mainly by endothelial cells and fibroblasts in our data, known to promote CRC invasion (*Drev et al., 2019*); and *Ctss*, a peptidase expressed by T cells and monocytes-macrophages that promotes CRC neovascularization and tumor growth (*Burden et al., 2009*). Finally, gene expression patterns in endothelial cells and monocytes in region 6 suggested active angiogenesis through a multi-cellular feedback loop, with enriched numbers of vascular and lymphatic endothelial cells expressing immune-attracting chemokines (*Cxcl9*) and adhesion molecules (e.g. *Chd5, Mcam*), monocytes expressing proangiogenic factors that induce proliferation of endothelial cells (e.g. *Mmp12*), and monocytes and macrophages expressing *Ctsd*, which increases tumorigenesis in CRC models (*Basu et al., 2019*; *Figure 5—figure supplement 1E*).

Region 8 was enriched for deep crypt cells (program 13; FDR = 1.7 10$^{-14}$, two-sided Welch's t-test on CLR transformed compositions), reminiscent of the normal stem cell niche in normal colon, an epithelial innate immune program (program 1; FDR = 3.4 10$^{-100}$) expressed by secretory cells in AV and AKPV lesions, and plasma and B cell activity. Unlike the canonical (normal) deep crypt region (region 5), which is enriched for MHCII expression (program 18; FDR = 8.6 10$^{-28}$), this region was depleted for the program's expression (FDR = 2.1 10$^{-26}$), which may indicate an earlier stem cell-like state (*Biton et al., 2018*), or a decoupling of the cell cycle and MHCII programs (which are coupled in normal ISC differentiation, and allow a cross talk with T cells to modulate T cell differentiation) (*Figure 5B and C*). The region's non-epithelial compartment was enriched for B cell activation and BCR signaling genes (FDR = 4.0 10$^{-4}$, 1.9 10$^{-3}$, two-sided Fisher's exact test in GO term enrichment, *Supplementary file 2*). This may be related to B cell function in protection from lumen antigens (*Spencer and Sollid, 2016*) or to tertiary lymphoid structures (TLS), which are correlated with clinical benefits in cancer patients (*Sautès-Fridman et al., 2019*). Notably, Epi05 (dysplastic secretory-like) enriched in Region 8 (*Figure 5B*) expressed higher levels of inflammatory genes and immune chemokines (e.g. *Ccl9, Ifitm3*) compared to normal counterparts, Epi04 (secretory; *Figure 3C*, *Figure 3—figure supplement 1C and E*), and may thus promote the formation of this region. We validated the presence of TLS-like structures in association with deep crypt secretory cells in AV lesions using multiplex RNA analysis (Cartana; *Figure 5—figure supplement 2*), showing that the dominant population of B cells is accompanied by monocyte-macrophages and T cells characteristic of TLSs, as well as the expression of *Reg4* and *Muc2*, deep crypt goblet/secretory cell markers.

Region 11 was populated by cells expressing the Wnt signaling pathway program (4, FDR = 9.3 10$^{-13}$, two-sided Welch's t-test on CLR transformed compositions), with several lines of evidence supporting an active epithelial to mesenchymal transition (EMT) in this region. Epithelial cells in region 11 were enriched for the expression of mesenchymal genes, including Vimentin (*Mendez et al., 2010*) (*Vim*, FDR = 7.4 10$^{-245}$, one-sided Fisher's exact test), *Prox1* (*Lu et al., 2012*) (FDR = 3.7 10$^{-153}$), and Sox11 (*Oliemuller et al., 2020*) (FDR = 7.7 10$^{-224}$) (*Figure 5—figure supplement 1F*), as well as for EMT signatures from a mouse model of lung adenocarcinoma (*Marjanovic et al., 2020*) (FDR = 1.5 10$^{-142}$, two-sided Mann-Whitney U test) and from human head and neck squamous cell carcinoma tumors (*Puram et al., 2017*) (FDR = 6.9 10$^{-55}$, two-sided Mann-Whitney U test). This is consistent with the role of Wnt signaling in promoting EMT and a mesenchymal phenotype in CRC, breast cancer, and other epithelial tumors (*Schwab et al., 2018*; *DiMeo et al., 2009*). Region 11 non-epithelial cells also expressed genes encoding MHC-I binding proteins (FDR = 4.6 10$^{-2}$, two-sided Fisher's exact test in GO term enrichment, *Supplementary file 2*) and actin cytoskeleton, filament, and binding proteins (FDR = 3.1 10$^{-6}$, 7.7 10$^{-3}$, 2.7 10$^{-10}$). Organization of the cytoskeleton affects migration, adherence, and interaction of lymphocytes with antigen-presenting cells (*Penninger and Crabtree, 1999*). Notably, region 11 also concentrated at a more distal part of the tissue at ~900 µm from the muscularis, suggesting an outgrowth of the tissue towards the lumen (*Figure 5—figure supplement 1C*).

Non-epithelial cells formed two cellular hubs in the malignant-like regions (6, 8, and 11; *Figure 5—figure supplement 1G*): An endothelial-fibroblast hub, detected in all three regions, and an immune hub with B cells, TNK cells, monocytes, and macrophages, which was prominent in inflammatory region 6, weaker (less spatially correlated) in region 8 (but validated in situ), and not correlated in region 11. Thus, activation of an immune response is reflected by close proximity between immune cells. We further characterized the organization of the vascular niche using our multiplex in situ RNA data, finding that while neighbors of the Pdgfrb-expressing pericytes are mainly other Pdgfrb-expressing pericytes and endothelial cells, Pecam1-expressing endothelial cells appear self-enriched next to themselves at cellular scale distances and close to Pdgfrb-expressing pericytes for larger distances (*Figure 5—figure supplement 1H*).

Overall, three multicellular community regions were enriched in AV lesions: (1) inflammatory epithelial regions with endothelial cells and monocytes expressing angiogenesis, inflammation, and invasion programs; (2) epithelial stem-like regions, associated with plasma and B cell activity; and (3) regions with epithelial to mesenchymal transition (EMT) and Wnt signaling dysplastic cells. Each region highlights different processes that modulate tumorigenesis or invasion, and the three regions co-exist in the same tumor at different spatial locations.

## Cell-cell interactions are rewired in AV lesions

To identify cell-cell signaling mechanisms that may underlie these regional associations, we used COMMOT (*Cang et al., 2023*), a computational framework that uses Optimal Transport to infer cell-cell communication from receptor-ligand expression patterns in spatially resolved data. We used COMMOT's bead-wise communication 'output' and devised a method to address p-value inflation in statistical enrichment testing, using spatially-informed data aggregation (Methods).

We observed stronger and distinct ligand-receptor interactions in AV *vs.* normal samples, reflecting the activated state in the dysplastic tissue (*Figure 5K*, *Figure 5—figure supplement 3A*). In particular, while interactions enriched in AV lesions involved immune, epithelial, and stromal signaling, those enriched in normal tissue involved neuropeptides, such as NPY and GCG (glucagon). Moreover, malignant-like regions 6 and 11 as well as region 2 (muscularis) were particularly enriched for active communication pathways (*Figure 5—figure supplements 1I and 3B*). This included the WNT signaling pathway, angiogenesis (VEGF, PDGF, FGF), and the OSM pathway.

## Similar spatial patterns in human and mouse tumors

The overall spatial distribution of cell types and epithelial profiles was conserved between mouse and human tumors, when comparing to scRNA-seq (*Pelka et al., 2021*; *Chen et al., 2021*; *Becker et al., 2022*; *Che et al., 2021*; *Zheng et al., 2022*; *Khaliq et al., 2022*; *Joanito et al., 2022*). We examined mouse-defined regions in human tumors, using TACCO to map the expression profiles associated with the epithelial, immune, and stroma compartments in each of the TACCO-identified mouse regions to scRNA-seq profiles from human CRC, and probabilistically annotated region-specific expression profiles for each scRNA-seq profile from the human samples. This identified two main 'meta compartments', with epithelial, stromal, and immune profiles from human MMRp and MMRd tumors associated with regions 6, 8, and 11 that were enriched in AV lesions (as well as 0 and 2), while those from normal human tissue were associated with normal regions (e.g. 5, 10, 12; *Figure 6A*, *Figure 6—figure supplement 1A*, Methods).

## Malignant-like regions are associated with tumor progression in human colorectal tumors

We next assessed if the regional epithelial programs that we spatially identified in mouse are conserved in human. To this end, we constructed pseudo-bulk profiles from epithelial cells for our mouse samples and for recently published human samples profiled along different stages of malignant transformation, from normal tissue to polyp to CRC (*Pelka et al., 2021*; *Chen et al., 2021*; *Becker et al., 2022*; *Che et al., 2021*; *Zheng et al., 2022*; *Khaliq et al., 2022*; *Joanito et al., 2022*). We scored each epithelial pseudo-bulk profile with the differentially expressed genes between the epithelial parts of the regions and computed the principal components of these scores across all human and mouse samples (Methods). The first principal component (PC1) captured features that are related to malignancy, with higher values for human tumors *vs.* polyps (*Figure 6B*, *Figure 6—figure supplement*

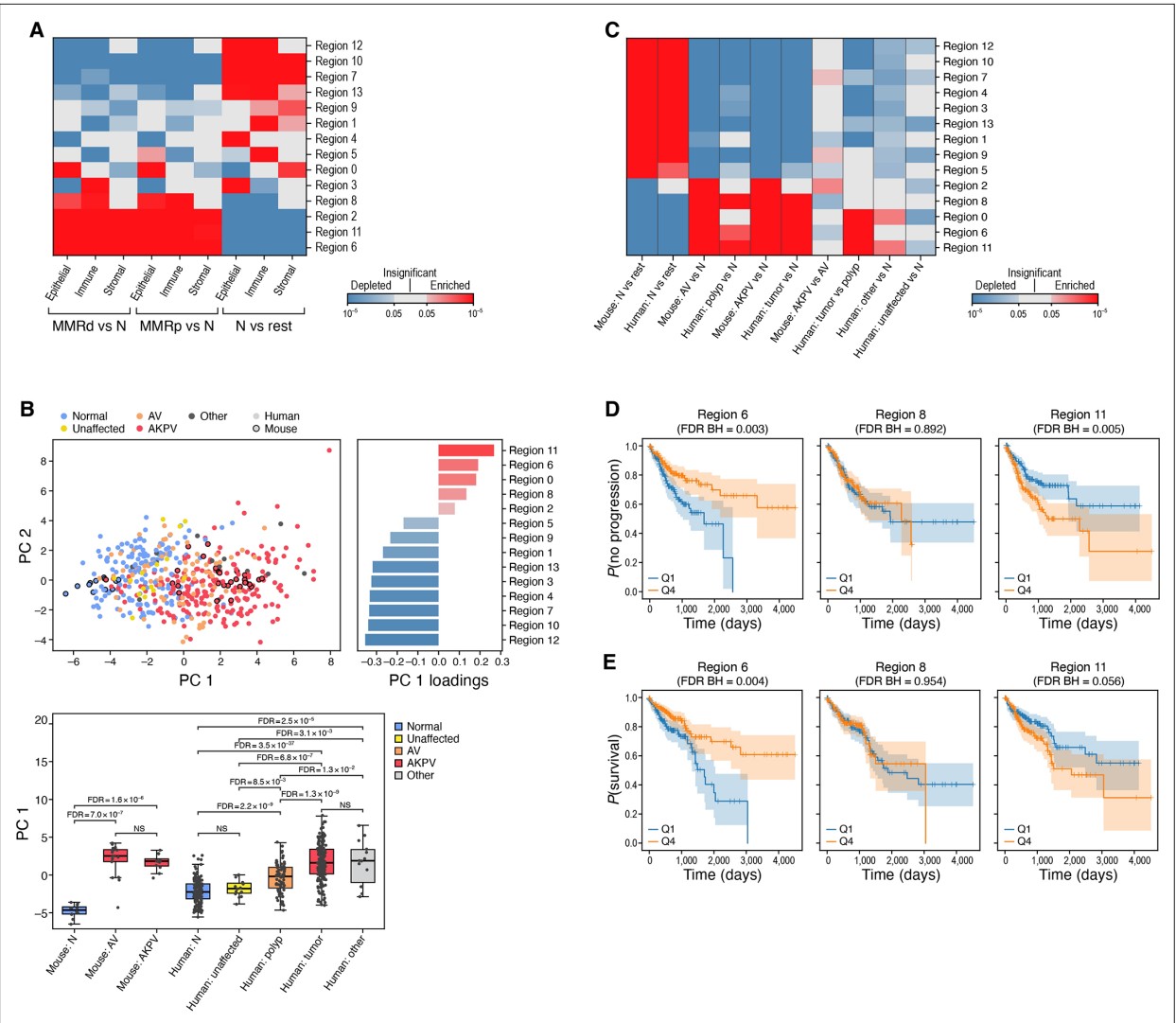

**Figure 6.** Mouse tumor regions associated with tumor progression in human colorectal tumors. (**A**) Expression profiles characterizing mouse regions are recapitulated in human tumors. Significance (FDR, color bar, two-sided Welch's t-test on CLR-transformed compositions) of enrichment (red) or depletion (blue) of region-associated epithelial, immune, or stromal profiles (rows) compared between normal, MMRp, or MMRd samples (columns). (**B, C**) Mouse regions capture malignant features in human tumors. (**B**) Top left: First (PC1, x axis) and second (PC2, y axis) principal components of mouse region scores of mouse and human epithelial pseudo-bulk samples. Top right: PC1 loadings (x axis) of each mouse region score (y axis). Bottom: PC1 values (box plots show mean, quartiles, and whiskers for the full data distribution except for outliers outside 1.5 times the interquartile range (IQR)) for each type of mouse or human sample (x axis). (**C**) Significance (FDR, color bar, two-sided Welch's t-test) of enrichment (red) or depletion (blue) of region-associated profile scores (rows) in normal and dysplastic samples (columns) in human or mouse. (**D, E**) Expression of malignant-like regions 6 and 11 in tumors is associated with PFI (**D**) and OS (**E**) in human patients. Kaplan-Meier PFI (**D**, n=662 *Liu et al., 2018*) or OS (**E**, n=662 *Liu et al., 2018*) analysis of human bulk RNA-seq cohort stratified by malignant-like region profile scores.

The online version of this article includes the following figure supplement(s) for figure 6:

**Figure supplement 1.** Transfer of mouse spatial region expression profiles to human patient data.

*1B*). In addition, malignant-like region (6/8/11) scores were higher in dysplastic *vs.* normal samples (*Figure 6C*). Thus, the spatial region profiles defined in mouse capture features that correlate with malignant transformation in humans.

We further classified each full pseudo-bulk profile from the dysplastic human samples into one of the four groups in the CMS expression-based classification (*Guinney et al., 2015*; Methods) and compared the mouse region scores for each class of samples (*Figure 6—figure supplement 1C*, Methods). CMS2 classified samples were most closely related to our dysplastic mouse models: all

of the dysplasia-associated regions were enriched in CMS2 tumors while most normal regions were depleted, relative to the other CMS classes.

Finally, we found that expression of the malignant-like regions (6/8/11) in tumors was associated with clinical outcome. We scored each tumor based on genes that were differentially expressed between the full expression profile of malignant-like regions (6/8/11) and compared the PFI and OS for patients in TCGA whose RNA-seq profiles were in the top and bottom quartile of malignant-like region scores (Methods). High scores for malignant-like region 11 (EMT) were correlated with shorter PFI, while those for malignant-like region 6 (inflammation) correlated with longer PFI and longer OS (*Figure 6D and E*). These associations were driven primarily by MMRp tumors (*Figure 6—figure supplement 1D and E*). This suggests that region 11 is associated with pro-tumorigenic properties in human patients, while region 6 might be associated with tumor-controlling properties. This highlights the importance of multicellular functional tissue modules in the CRC tumor ecosystem.

## Discussion

Here, we systematically charted the spatial organization of cellular expression in dysplastic tissue of the colon to help identify putative functional units in the tumor. We used TACCO (*Mages et al., 2023*) to integrate scRNA-seq and Slide-seq data, not only by mapping cell types to their positions, but also distinguishing different cell programs, the regions that they dominate, and their characteristic microenvironments. This allowed us to overcome technical limitations, such as lack of spatial context in scRNA-seq and sparse readout in Slide-seq, and to generate a high-resolution spatial map of the dysplastic landscape transcending beyond the mapping of individual cells to spatial positions. We used this map to show correlation with clinical outcome in human patient tumors.

Our scRNA-seq analysis revealed profound enrichment of a stem cell program in dysplastic tissues. The profiles of dysplastic cells expressing this program are distinct from normal stem cells and enriched with expression of negative regulators of the WNT signaling pathway and inflammation, suggesting a non-canonical function. The abundance of these cells with stemness potential across all our malignant-like regions points to a dynamic population that can affect the cells in its proximity by secretion of negative regulators of the WNT signaling and inflammatory function but may also adopt various functions depending on the environmental cues and dysplasia-associated cells in its proximity. A similar population, designated 'high-plasticity cell state', was previously described in a mouse model of lung adenocarcinoma and in human patients, where it was correlated with resistance to chemotherapy (*Marjanovic et al., 2020*). Whether these cells can be manipulated to take on specific phenotypes or even to differentiate into normal-like enterocytes given the appropriate signal from the microenvironment remains as open questions.

Within the dysplastic lesions, alongside malignant-like regions, we found regions with normal features (regions 3, 4, 9, and 10), comparable to regions found in the normal colon, most likely representing compartments driven by clones that were not affected by the genetic perturbation. One of these regions, region 4, contained mainly goblet cells with normal expression profiles. Whether this neighborhood represents normal cells that reside alongside malignant cells or a cancer transition state, it may modify tumor progression by recruiting immune cells or by secreting factors that affect epithelial proliferation in adjacent regions. For example, region 4 in dysplastic lesions is enriched with chemokine activity genes relative to region 4 in normal colon, suggesting a possible role in recruitment of immune cells to the dysplastic landscape. Further work is required to understand the role of these regions (expressing normal features) in tumor progression.

While the malignant-like regions were identified as discrete spatial entities, each with coordinated features across epithelial, immune, and stromal cells, these regions are adjacent to each other. Thus, they may still influence one another through signaling or by utilizing branches of the same main vessels. For example, *Osm* is expressed by cells in region 6, whereas its receptor is expressed on fibroblasts and endothelial cells enriched in region 11. *OSMR* was previously shown to be expressed by inflammatory fibroblasts (*Smillie et al., 2019*; *West et al., 2017*) and its activation in malignant cells promotes EMT in breast cancer and pancreatic cancer (*West et al., 2014*; *Smigiel et al., 2017*) and a mesenchymal state in glioblastoma (*Hara et al., 2021*). Future studies can help determine if these regions are functionally interdependent and if they evolved from the same clones and can interconvert, or whether they developed independently.

Because animal models complement cell and tissue atlases of human CRC (*Pelka et al., 2021*; *Chen et al., 2021*), by allowing experimental manipulation for mechanistic studies (*de Sousa e Melo et al., 2017*; *Schepers et al., 2012*), it is important to relate between models and patients. By studying genetically engineered mouse models using high-resolution single-cell spatial genomics, we can help determine to what extent they recapitulate the cellular and spatial organization of human disease, in the context of two distinct genetic states that represent human CMS-2 lesions. To this end, we developed several approaches to allow cross-species comparison of tumors at the single-cell and spatial level, despite the high level of both intra- and inter-individual variation within each species. Comparing to human CRC, our analysis suggests that the CRC landscape is organized in similar multicellular functional tissue modules between human and mouse, and disease subtypes (e.g. MMRp and MMRd). Future studies applying our approaches to patient cohorts could help understand whether the expression of different tissue modules may contribute to the partial response to immunotherapy reported for MMRd patients (*André et al., 2020*), and to define specific tissue modules predictive of response to therapy. Notably, while our study focused on the tumor landscape, its findings may be relevant for tissue response to other challenges (e.g. inflammation, fibrosis, wound healing), which involve activation of similar functional tissue modules, a result of collective function of parenchymal, immune, and stromal cells.

Taken together, our integrative approach facilitates spatial analysis with high resolution, constructing regional neighborhoods and their spatial layout at both high cellular resolution and genomic scale. Our work is an important step toward a systematic understanding of the organization of dysplastic tissue with the potential to contribute to improved patient stratification by the multicellular functional units in the tumor landscape.

## Materials and methods

### Mice

Mice were housed in the animal facility at the Koch Institute for Integrative Cancer Research at MIT. All animal studies described in this study were approved by the MIT Institutional Animal Care and Use Committee (Protocol 1213-106-16). $Apc^{fl/fl}$ mice (*Kuraguchi et al., 2006*) were obtained from NCI mouse repository; $Kras^{LSL-G12D/+}$ (*Johnson et al., 2001*), $Rosa26^{LSL-tdTomato}$ (*Madisen et al., 2010*), and $Trp53^{fl/fl}$ (*Marino et al., 2000*) mice obtained from Jackson; and $Villin^{CreERT2}$ (*el Marjou et al., 2004*) mice were a gift from Dr. Sylvie Robine. All mice were maintained on C57BL/6 J genetic background. Approximately equal numbers of male and female mice of 6–10 weeks of age were used for all experiments. Where indicated, mice were injected to the submucosal layer of the colon with 4-hydroxytamoxifen (EMD Millipore # 579002) dissolved in ethanol at a concentration of 100 µM (for the mice that were kept for 3 weeks after injection) or 30 µM (for the mice that were kept for 9 weeks after injection). Tumors were resected at either 3 or 9 weeks after 4-hydroxytamoxifen injection. Colonoscopy and colonoscopy-guided injection methods were previously described in detail (*Roper et al., 2017*; *Roper et al., 2018*).

### Tissue processing for scRNA-seq

Single-cell suspensions from healthy colon or dysplastic lesions were processed using a modified version of a previously published protocol (*Smillie et al., 2019*). Tissue samples were rinsed in 30 ml of ice-cold PBS (Thermo Fisher 10010–049), chopped to small pieces, and washed twice in 25 ml PBS, 5 mM EDTA (Thermo Fisher AM9261), 1% FBS (Thermo Fisher 10082–147). To prime tissue for enzymatic digestion, samples were incubated for 10 min at 37 °C, placed on ice for 10 min before shaking vigorously 15 times followed by supernatant removal. Tissues were placed into a large volume of ice-cold PBS to rinse prior to transferring to 5 ml of enzymatic digestion mix (Base: RPMI1640, 10 mM HEPES (Thermo Fisher 15630–080), 2% FBS), freshly supplemented immediately before use with 100 mg/ml of Liberase TM (Roche 5401127001) and 50 mg/ml of DNase I (Roche 10104159001), and incubated at 37 °C with 120 rpm rotation for 30 min. After 30 min, enzymatic dissociation was quenched by addition of 1 ml of 100% FBS and 10 mM EDTA. Samples were then filtered through a 40 mM cell strainer into a new 50-mL conical tube and rinsed with PBS to 30 mL total volume. Tubes were spun down at 400 $g$ for 7 min, at 4 °C. Resulting cell pellets were resuspended in 1 ml PBS, placed on ice, and counted.

## Cell hashing

Cell hashing was performed based on the published protocol (*Stoeckius et al., 2018*) as summarized below. Dissociated cells were resuspended in 1 ml of Cell Hashing Staining Buffer 1×PBS with 2% BSA (New England Biolabs, B9000S) and 0.02% Tween (Tween–20 Solution, 10%, Teknova, VWR-100216–360) and counted. 500,000 cells were resuspended in 100 µl of Cell Hashing Staining Buffer and incubated for 30 min on ice, with 2 µl of the appropriate BioLegend TotalSeq Hashing antibody (a 1:50 dilution, using a total of 1 µg of antibody per cell suspension). TotalSeq-A anti-mouse Hashtag antibodies #1–8 (catalog numbers: 155801, 155803, 155805, 155807, 155809, 155811, 155813, 155815) were used. Cells were washed three times with 0.5 ml of Cell Hashing Staining Buffer and filtered through low-volume 40-µm cell strainers. All cell suspensions were recounted to achieve a uniform concentration of 7000 cells per microliter before pooling for capture by 10x Chromium controller following the manufacturer protocol for the v2 or v3 3' kit (10x Genomics, Pleasanton, CA).

## Hashtag oligo (HTO) library preparation

Separation of hashtag oligo (HTO)-derived cDNAs (<180 bp) and mRNA-derived cDNAs (>300 bp) was done after whole-transcriptome amplification by performing 0.6×SPRI bead purification (Agencourt) on cDNA reactions as described in 10x Genomics protocol. Briefly, supernatant from 0.6×SPRI purification contains the HTO fraction, which was subsequently purified using 1.4 and 2×SPRI purifications per the manufacturer's protocol (Agencourt). HTOs were eluted by resuspending SPRI beads in 15 µl TE. Purified HTO sequencing libraries were then amplified by PCR (1 µl clean HTO cDNA, 25 µl 2 X NEBNext Master Mix [NEB #M0541]), 10 µM SI-PCR and D701 or D704 primers performed dial-out PCR (98 °C (10 s), (98 °C for 2 s, 72 °C for 15 s) x 12/18 then 72 °C for 1 min) for 12 and 18 cycles, and used the 18 cycles product for sequencing. PCR reactions were purified using another 2×SPRI clean up and eluted in 15 µl of 1×TE. HTO libraries were quantified by Qubit High sensitivity DNA assay (ThermoFisher) and loaded onto a BioAnalyzer high sensitivity DNA chip (Agilent).

SI-PCR: AATGATACGGCGACCACCGAGATCTACACTCTTTCCCTACACGACGC*T*C
D701: CAAGCAGAAGACGGCATACGAGATCGAGTAATGTGACTGGAGTTCAGACGTGTGC
D704: CAAGCAGAAGACGGCATACGAGATGGAATCTCGTGACTGGAGTTCAGACGTGTGC

## Sequencing

Samples were sequenced using HiSeq X (Illumina). Hashing libraries were sequenced with spike-ins of 2.5%.

## Tissue processing for Slide-seq

Colons were flashed with cold PBS and a segment that includes the lesion and surrounding tissue (or a respective healthy segment from normal mice) was dissected. Samples were then mounted in cold OCT, flash frozen on dry ice covered with ETOH, and long-term stored in –80°C.

## Slide-seq

For mouse and human experiments, 10 µm sections were cut and the Slide-seq V2 protocol was used as previously described (*Stickels et al., 2021*). For mouse experiments, four and six arrays were collected from normal colons and AV lesions, respectively. The muscularis was fit onto the array of both healthy and dysplastic lesions to allow appropriate orientation.

## Multiplex in situ RNA analysis

Multiplex in situ RNA analysis was performed with Cartana (*Gyllborg et al., 2020*) technology (a newer version is now available as Xenium [10x Genomics]). In total, we measured three samples with one section per sample in each state (normal/AV) and probe set (V1/V2), with an additional replicate section for one of the samples in normal V2.

Fresh Frozen OCT-embedded tissues from normal colon and AV lesions were cryosectioned as 10 µm sections and placed onto SuperFrost Plus glass slides (Thermo Fisher) and further stored at –80 °C before experiments. Samples were fixed (with 4% formaldehyde) for 5 min and permeabilized for 5 min (with 0.1 mg/ml pepsin in 0.1 M HCl [P7012 Sigma-Aldrich]) prior to library preparation.

For library preparation, chimeric padlock probes (directly targeting RNA and containing an anchor sequence as well as a gene-specific barcode) for a custom panel of 66 (V1) or 180 (V2) genes (*Supplementary file 3*, see below) were hybridized overnight at 37 °C, then ligated before the rolling circle amplification was performed overnight at 30 °C using the HS Library Preparation kit for CARTANA technology and following manufacturer's instructions. All incubations were performed in SecureSeal chambers (Grace Biolabs). Note that prior to final library preparation, optimal RNA integrity and assay conditions were assessed using Malat1 and Rplp0 housekeeping genes only using the same protocol.

To prevent tissue sections from detachment, an additional baking step of 30 min at 37 °C was performed before mounting. To quench autofluorescence background, TrueView (SP-8400 Vector-Labs) was used for 1 min at room temperature. For tissue sections mounting, Slow Fade Antifade Mountant (Thermo Fisher) was used for optimal handling and imaging.

Quality control of library preparation was performed by applying anchor probes to detect simultaneously all rolling circle amplification products from all genes in all panels. Anchor probes are labeled probes with Cy5 fluorophore (excitation at 650 nm and emission at 670 nm).

All samples passed quality control and went through in situ barcode sequencing, imaging, and data processing. Briefly, adapter probes and sequencing pools (containing four different fluorescent labels: Alexa Fluor 488, Cy3, Cy5, and Alexa Fluor 750) were hybridized to the padlock probes to detect the gene-specific barcodes, through a sequence-specific signal for each gene-specific rolling circle amplification product. This was followed by imaging and performed six times in a row to allow for the decoding of all genes in the panel. To reduce lipofuscin autofluorescence, 1 X Lipofuscin Autofluorescence Quencher (Promocell) was applied for 30 s before fluorescence labeling.

Raw data consisting of 20x or 40x images from five fluorescent channels (DAPI, Alexa Fluor 488, Cy3, Cy5, and Alexa Fluor 750) were each taken as z-stack and flattened to 2D using maximum intensity projection. After image processing and decoding, the results were summarized in a csv file and gene plots were generated using MATLAB (*Qian et al., 2020*).

## scRNA-seq pre-processing and quality control filtering

Count matrices for scRNA-seq were generated using the Cumulus feature barcoding workflow v0.2.0 (*Li et al., 2020*) with CellRanger v3.1.0 and the mm10_v3.0.0 mouse genome reference. Cell profiles were quality filtered by requiring between 1000 and 50,000 counts, and between 500 and 7000 genes, less than 20% mitochondrial counts, and less than 10% hemoglobin counts. Cell profiles that did not meet all these criteria were discarded. The top 5000 highly variable genes were annotated on the remaining cells after normalization to 10,000 counts and log1p transform using Scanpy's 'highly_variable_genes' function (*Wolf et al., 2018*) and providing the chemistry (v2/v3) by hashing (True/False) combination as batch-annotation. Putative doublets were removed using Scrublet (*Wolock et al., 2019*) with default parameters.

## Selection of variable genes, dimensionality reduction, and clustering

A preliminary clustering using the Leiden algorithm with resolution 1.0 was performed after normalization to 10,000 counts, log1p transform, correction for number of counts and percentage of mitochondrial genes, scaling with a max_value of 10, and generating a $k$-nearest neighbors ($k$-NN) graph with 15 neighbors on a PCA of the previously annotated 5000 highly variable genes with 50 components using Scanpy (*Wolf et al., 2018*). The single-cell profiles were provisionally annotated with SingleR (*Aran et al., 2019*) cell-wise (i.e. without using clustering information) using the SingleR built-in MouseRNAseqData and an intestine-specific dataset from Tabula Muris (*The Tabula Muris Consortium et al., 2018*; https://figshare.com/ndownloader/files/13092143). For further processing, the dataset was then split into the three compartments, epithelial, immune, and stromal, using the provisional SingleR annotations.

For each compartment, the top 5000 highly variable genes were annotated using Scanpy's 'highly_variable_genes' function on cells normalized to 10,000 counts after log1p-transformation and providing the chemistry (v2/v3) by hashing (True/False) combination as batch-annotation.

## Expression programs and batch correction

For the dataset of each compartment separately (generated as described above), an integrative NMF was performed (using a part of the LIGER *Welch et al., 2019* implementation) with k=20 and

lambda=5 to identify 20 programs and their respective weights per cell. This iNMF factorization represents the single cell expression matrix as a weighted sum of profiles such that both the weights and programs contain only non-negative numbers, while allowing for and separating out batch-only contributions. The same approach was also used with a higher $k$ (epithelial and immune: 200, stromal: 50) to yield a detailed and batch-corrected decomposition of expression which was then combined to obtain a count-like corrected expression matrix for the top 5000 highly variable genes. For each compartment separately, these batch-corrected data were normalized to 10,000 counts, log1p transformed, corrected for number of counts and percentage of mitochondrial genes by linear regression, scaled with a max_value of 10, followed by a PCA of the previously annotated 5000 highly variable genes. A $k$-nearest neighbors ($k$-NN) graph was constructed from the top 50 PCs, with $k$=15 neighbors using Scanpy, and clustered using a compartment-specific Leiden resolution parameter (epithelial: 0.2, immune: 0.4, stromal: 0.1). This clustering was used as the cluster level annotation of the mouse scRNA-seq data for the epithelial and stromal compartment. Separately per compartment, the data were annotated with SingleR using the cluster information. The same per-compartment batch-corrected and preprocessed data from the Leiden clustering was used to create UMAP embeddings with PAGA initialization using Scanpy.

To improve the clustering and annotation in the immune compartment and to filter out additional doublets not detected by Scrublet, the immune data were separately filtered and clustered using information from the compartment level clustering and annotation. To that end, myeloid and T/NK cells were partitioned separately and further processed, and additional likely doublet cells were labeled and removed by the following procedure:

1. Cells were labeled as doublets based on higher number of UMIs of marker genes for other compartments than the 95th percentile observed in this immune partition (i.e. Epcam and Cdh1 to remove immune-epithelial doublets and Cav1 and Kdr to remove immune-stromal doublets) and other immune partitions (i.e. Cd3d, Cd3e, and Cd3g to remove myeloid-lymphoid doublets from the myeloid cells). This type of filter criterion for lowly expressed genes ('larger than some percentile' on integer counts) also allows keeping more than 95% of the cells if, for example, all cells of this partition happened to have 0 UMIs of a particular marker gene.
2. Cells were labeled as doublets if they had inconsistent cell-wise and cluster-wise SingleR annotations.
3. Cells were labeled as doublets if they had significantly (Benjamini-Hochberg FDR = 0.05, one-sided Fisher's exact test) more neighbors in the k-NN graph from the immune compartment that were already marked as doublets.
4. All cells labeled as doublets were removed.

After filtering, the count matrices were batch corrected as above using the integrative NMF from LIGER with k=20 and lambda = 5, and clustered like above with group-specific Leiden resolution (myeloid: 0.2, TNK: 0.4). For myeloid and TNK cells, this clustering superseded the original clustering. The integrative NMF result here was only used for updating the clustering and not for generating an extra set of expression programs.

Note that cluster Epi06 shows a broad expression spectrum; while such pattern can best be explained by remaining doublets, they were not called by Scrublet and could also not be consistently removed by extra QC as was applied to the immune compartment. The interpretation of Epi06 as containing doublets is supported by its overrepresentation in the annotation of the Slide-seq data (*Figure 4—figure supplement 1E*), as both doublets and Slide-seq beads represent compositions of different cells. Still, non-doublet explanations (e.g. undifferentiated cells) cannot be ruled out. Therefore, Epi06 cells are retained in the data but not interpreted biologically.

## Marker selection for in situ RNA analysis (CARTANA)

CARTANA V1 markers were selected from genes differentially expressed between compartments, cell types, and clusters (for the immune and stromal compartments), and highly ranked genes for programs (for the epithelial compartment) were first filtered by biological relevance and the literature, obtaining 87 genes. To further reduce the set to the available panel size (66 genes), genes were annotated into categories of potential redundancy. To choose between redundant genes, a global objective function was optimized over the gene selection, looping over all potentially redundant gene sets until convergence, exhaustively testing all choices within a gene set, and accepting the best choice for

this gene set in terms of the global objective function. The global objective function was constructed as the mean fourfold cross-validation scores (using 'GroupKFold' and 'cross_val_score' from sklearn *Pedregosa et al., 2011*) of multi-class logistic regression classification (using 'LogisticRegression' from sklearn) for discriminating cell classes and of ridge regression of epithelial program weights (using 'RidgeRegression' from sklearn) on the scRNA-seq data subsampled to the expected sparsity of CARTANA data. The cell classifications used were between the stromal, immune, and epithelial compartment, within the stromal compartment between Endo01, Endo02, and fibroblasts, within the immune compartment between myeloid and lymphoid lineage, within the lymphoid lineage between T cells and B cells, within the myeloid lineage between granulocytes, mast cells, and all monocytes and macrophages together, between monocytes and macrophages, and within monocytes between Mono01, Mono02, Mono03, and Mono04. The programs used in ridge regression were programs 3, 4, 6, 7, 13, 14, 15, and 16. As the probes for Ly6c1 and Ly6c2 could not discriminate sufficiently between Ly6c1 and Ly6c2, we chose combined probes that measure both.

The CARTANA V2 panel included 59 of the 66 genes in the V1 panel (the others had to be removed for technical reasons), another 113 genes from the standard fixed gene panel for CARTANA, and 8 selected genes from literature.

## Analysis of CARTANA data

Each measured molecule was annotated with an originating cell type cluster label (using TACCO's 'tc.tl.annotate_single_molecules', with RCTD *Cable et al., 2022* as the core annotation method and parameters bin_size = 20, n_shifts = 3, assume_valid_counts = True) separately for each sample. For this, genes in the reference that would likely cross-hybridize in the probe panel design were summed over (Ly6c1 and Ly6c2). TLS-like regions were annotated by visual inspection of the cell type cluster composition and morphology.

To assess cell type compositions of the full dataset, molecules with cell type cluster annotations were binned into 10 μm bins (using TACCO's 'tc.utils.bin' and 'tc.utils.hash' functions) and cluster-level annotations were merged to cell-type level.

To assess the compositions of TLS-like regions, CARTANA v2 data were aggregated, conserving the categorical TLS annotation (using TACCO's 'tc.utils.bin' and 'tc.tl.dataframe2anndata' functions).

## Comparison between experimental methods

To compare cell type composition between methods, CLR-transformed compositions of samples (or of spatially split samples for the spatial methods, see subsection 'Ligand-receptor analysis and spatially informed enrichment') were computed. Then, using a 100 bootstrapped means of the compositions, mean and standard error of the mean were calculated once for the AV and once for the normal samples. To compare gene expression, the mean difference of gene counts between normal and AV samples and its standard error were calculated using all pairwise differences between the bootstrapped normal and AV samples. For gene comparisons, the CLR-transformed composition over all genes that were measured in all of scRNA-seq, Slide-seq, CARTANA V1, and CARTANA V2 was used. Pearson correlation between mean compositions was calculated for each pair of methods.

## RNA-velocity analysis

Splicing-aware count matrices for scRNA-seq were generated using CellRanger v6.1.2 and velocyto v0.17.17 (*La Manno et al., 2018*) with the ensembl v108 mouse genome reference. Scvelo v0.2.5 was used to infer velocity separately for the epithelial and TNK subsets (using the functions 'scv.pp.filter_and_normalize', 'scv.pp.moments', 'scv.tl.velocity' (with mode='stochastic'), and 'scv.tl.velocity_graph'). Scanpy and bbknn v1.5.1 (*Polański et al., 2020*) were used to generate batch-corrected UMAP embeddings for the two subsets for visualization with scvelo's 'scv.pl.velocity_embedding_stream' function.

## Selection of human single-cell data for the comparison of cell type and epithelial program composition

ScRNA-seq data from *Pelka et al., 2021* was used as reference for human CRC. To avoid biases in cell type compositions, only the subset of the data where 'PROCESSING_TYPE = unsorted' was used.

## Comparison of human and mouse samples by cell type composition

To compare human and mouse samples by composition of T/NK cell subsets, T/NK annotations from mouse and human data (*Pelka et al., 2021*) were matched by TACCO, using optimal transport (OT). First, human expression data were mapped to mouse genes using MGI homology information [subsection 'Mapping of mouse and human orthologs']. Then, human cell cluster annotations ('cl295v11Sub-Full') were mapped from the subset of human cells annotated as T/NK/ILC to the subset of mouse cells annotated as T/NK using TACCOs 'annotate' function with OT as core method, basic platform normalization, entropy regularization parameter epsilon 0.005, marginal relaxation parameter lambda of 0.1, and 4 iterations of bisectioning with a divisor of 3. Annotation with maximum probability per cell was used as the unique cluster level annotation for mouse T/NK cells. Annotations were aggregated per sample to yield a compositional annotation over the identical cluster annotation categories (from the human dataset) for the T/NK subsets of human and mouse samples. Annotation vectors were then processed using the sc.pp.neighbors and sc.tl.umap functions from Scanpy (*Wolf et al., 2018*) to yield a 2D sample embedding with respect to T/NK cell composition. Using the coordinates in the UMAP in place of spatial coordinates, neighborhood enrichment z-scores were computed with TACCO's co_occurrence_matrix function with max_distance = 2 and n_permutation = 100.

## Slide-seq compositional annotation

Slide-seq data were annotated with scRNA-seq reference annotations. First, Slide-seq and scRNA-seq data were filtered to retain only 15759 genes that were detected in both datasets and only beads and cells that had at least 10 counts across all these common genes.

Next, TACCO (*Mages et al., 2023*) was used to perform compositional annotation of each bead, allowing the bead counts to be explained by fractional contributions. In its basic application, TACCO finds an 'optimal' mapping between scRNA-seq annotation categories (e.g. cell types) and beads by solving a variant of an entropically regularized Optimal Transport (OT) problem in expression space. In its iterative application, TACCO uses a bi-sectioning functionality iteratively, annotating only fractions of the counts in each round and reserving the remainder for the next round to improve the sensitivity to sub-leading annotation contributions (i.e. first capture a portion of the counts for the 'top' cell types, but preserving others for other, more minor, cell types).

For the compositional annotation of Slide-seq beads with the categorical cell clusters from the single cell data, the 'annotate' function of TACCO with OT was used as core annotation method per Slide-seq puck with the subset of the single cell data with matching disease state, with basic platform normalization, entropy regularization parameter epsilon 0.005, marginal relaxation parameter lambda of 0.001, and four iterations of bisectioning with a divisor of 3.

For the compositional annotation of Slide-seq beads with the compositional epithelial programs, the annotated beads were split using the 'split_observations' function of TACCO on the cluster-level annotation and then aggregated to compartment level using the 'merge_observations' function keeping only beads for a compartment with at least 50 counts assigned to that compartment. The genes from the epithelial part were filtered to retain only those that were used to define the epithelial programs in scRNA-seq, and then annotated using again the 'annotate' function with OT as core annotation method, basic platform normalization, entropy regularization parameter epsilon 0.01, and a marginal relaxation parameter lambda of 0.001.

## Slide-seq region annotation

Region annotation was done for all pucks (normal and AV pucks) in one step to get comparable region annotations across pucks, such that beads that have similar epithelial program activity and are spatially close are called as one region. Because cell type composition can change drastically from one bead to its neighbor at the length scales of Slide-seq data, there is a need to compromise between optimizing the two similarities. This is done with the 'find_regions' function of TACCO, which performs a Leiden clustering on the weighted sum of connectivity matrices derived from epithelial program similarity and spatial proximity, using a position weight of 0.7, a Leiden resolution of 1.3, and 15 nearest neighbors per bead in position space and epithelial program space. To determine the neighbors in epithelial program space, the square roots of the program weights were used for neighbor finding, which effectively uses Bhattacharyya coefficients as overlap in epithelial program space instead of the Euclidean scalar products used for position space. These regions are defined by construction only on beads with

a large enough epithelial contribution (see above) and are then extended to all beads by assigning unannotated beads the region from the nearest bead with region annotation.

Submucosal and muscularis propria layers are predominantly comprised of fibroblasts and muscle cells, respectively, alongside blood and lymphatic vessels, nerves, and immune cells. Our algorithm depends on the epithelial expression component in beads. Since these layers do not contain epithelial cells (*Rao and Wang, 2010*), mapping likely relied either on 'noisy' signal from non-epithelial cells or from the basal-most epithelial layer.

To determine region composition at a certain distance of a reference region, TACCO's 'annotation_coordinate' function is used with max_distance = 1000 and delta_distance = 10.

## Slide-seq quality filtering

For all downstream analyses, all beads with less than 100 reads were discarded.

## Region- and cell type-characterizing genes in Slide-seq data

Genes to characterize regions on Slide-seq pucks irrespective of compartment composition were found using Scanpy's rank_genes_groups function on the full bead expression profiles. To find them separately for each compartment, the compartment-level split beads [sub-section 'Slide-seq annotation'] were used instead of the full beads. To compare gene expression between cell types on Slide-seq pucks, cluster-level split beads [sub-section 'Slide-seq annotation'] were aggregated to cell type level.

## EMT scoring

Malignant regions were scored for EMT signatures, using only counts attributed to the epithelial compartments within these regions and only genes expressed on at least three beads. Bead profiles were normalized to 10,000 counts, log1p transformed and scaled, and Scanpy's 'sc.tl.score_genes' function was used to score the top 50 genes in two EMT gene signatures (*Marjanovic et al., 2020*; *Puram et al., 2018*).

## Cell-type neighborships in Slide-seq data

To evaluate the local cell-type neighborship relations in the different disease states on the cluster level, the clusters were filtered per disease state to contain only clusters which account for at least 1% of the UMIs in that state. Then neighborhood-enrichment z-scores were calculated using TACCO's 'co_occurrence_matrix' function with max_distance = 20 and n_permutation = 10. To evaluate the stability of the result, this is also repeated for (max_distance, n_permutation) = (40,10), (60,10), (20,5), and (20,50). To get the significance of the overall change in z-scores between the states, a one-sided Mann-Whitney U test was performed on the values of the upper triangular half of the matrix between the two disease states for (max_distance, n_permutation)=(20,10).

A similar neighborship analysis was performed on the coarser cell-type level separately for the three malignant regions 6, 8, and 11, using TACCO's 'co_occurrence_matrix' function with max_distance = 20 and n_permutation = 10.

## Cell-type co-occurrence in Slide-seq data

Cell-type compositions relative to a spatial landmark, Region 2=muscularis, were evaluated using TACCO's 'annotation_coordinate' function with max_distance = 1000 and delta_distance = 10. To reduce tissue structure bias from the muscularis, the distance dependency of cell-type frequency relations was evaluated only for beads deep in the 'epithelial domain', defined as follows. The effective distance from stromal annotation was computed using TACCO's 'annotation_coordinate' function (with max_distance = 100, delta_distance = 10, critical_neighbourhood_size = 4.0) and only beads with a distance of at least 75 μm were used. On these remaining beads, TACCO's 'co_occurrence' function was used (with delta_distance = 20, max_distance = 1000) to compute cell types co-occurrence as a function of their distance.

## Epithelial program neighborships in Slide-seq data

As for cell types above, neighborship relations were evaluated for epithelial programs in the AV Slide-seq samples using TACCO's 'co_occurrence_matrix' function with max_distance = 20 and

n_permutation = 10, after selecting only the programs which make up at least 1% of the UMIs in the AV Slide-seq samples.

## Ligand-receptor analysis and spatially informed enrichment

For the ligand-receptor analysis on slide-seq data, we used COMMOT v0.03 (*Cang et al., 2023*). COMMOT employs optimal transport to construct sender and receiver side receptor-ligand interactions for every bead in one run of COMMOT. After filtering to beads with at least 100 counts, we applied basic preprocessing (normalization, log1p-transformation) and loaded the CellChat database as done in the COMMOT tutorial. We then follow the Slide-Seq v2 analysis from the COMMOT paper ('slideseqv2-mouse-hippocampus/1-lr_signaling.ipynb' from https://doi.org/10.5281/zenodo.7272562) to filter the database and to reconstruct the spatial communication network on ligand-receptor pair and pathway level separately for every puck. In particular, we use the distance cutoff of 200 µm for inference of ligand-receptor interactions. The resulting bead-wise sender- and receiver communication values were then used for enrichment analysis between disease states and between spatial regions.

Unlike the downstream Slide-Seq v2 analysis from the COMMOT paper ('slideseqv2-mouse-hippocampus/2-downstream_analysis.ipynb' from https://doi.org/10.5281/zenodo.7272562), we do not treat each bead on the puck as statistically independent observation in statistical tests, which leads to unreliably small p-values. Instead, we split all pucks along their axis of greatest extent (defined by the first principal component axis of the distribution of the spatial measurements) into spatial patches discarding 400 µm of boundary layer (twice the COMMOT distance cutoff, also co-occurrences have decayed strongly at this distance) in between the patches to reduce the correlation between the patches. This is done iteratively with the patches to get a set of weakly correlated patches. On these patches, we calculate the mean of the communication values and treat them as statistically independent observations for statistical tests. We argue that multiple sufficiently separated spatial patches of single spatial samples can be seen as multiple spatial samples using a spatial method with a smaller measurement area, and therefore can be treated as replicates. Unlike splitting into patches without removing a boundary layer, this procedure does not converge to the case of treating each bead as an independent observation as the number of iterations rises as it accounts for the spatial correlations between adjacent measurements. This gives a natural lower limit of p-values reachable with p-values rising again if too many splits are performed as too much data is lost to remove the correlations (*Figure 5—figure supplement 3C and D*). We chose a number of iterations of 2 as a compromise between having more patches and not discarding much data. For the enrichment of the pathway communication values, we used a two-sided Mann-Whitney U test across the patches and cite Benjamini-Hochberg FDR values. For the enrichment testing in a given group (e.g. an annotated spatial region), only those patches are used which have at least 100 beads on the patch.

Unless otherwise noted, for consistency, an analogous enrichment procedure was also used for COMMOT-unrelated quantities, like cell types, even though their spatial footprint is not as large as COMMOT's (distance cutoff of 200 µm). For cluster-level cell type enrichment analyses on Slide-seq data, only the top 3 contributing subtypes were considered, to better reflect the expected properties of Slide-seq data and better compare with categorically annotated scRNA-seq data.

## Mapping of mouse and human orthologs

We applied TACCO's functions 'setup_orthology_converter' and 'run_orthology_converter' with option 'use_synonyms = True' to map human to mouse genes using the ortholog mapping from Mouse Genome Informatics (http://www.informatics.jax.org/homology.shtml). Specifically, we used http://www.informatics.jax.org/downloads/reports/HOM_MouseHumanSequence.rpt (downloaded April 26th, 2021) for the analyses comparing T/NK compositions of human and mouse scRNA-seq data, analyses comparing epithelial program similarity in human and mouse scRNA-seq data, analyses involving the annotation of human scRNA-seq data with mouse regions, and analyses comparing the cell type and epithelial program correlations cross samples between human and mouse scRNA-seq data, and http://www.informatics.jax.org/downloads/reports/HOM_AllOrganism.rpt (downloaded on August 8th, 2022) for the analyses involving the scoring of mouse regions in mouse and human scRNA-seq data and leading to the CMS classification.

## GO term enrichment analysis

We used TACCO's functions 'setup_goa_analysis' and 'run_goa_analysis' to perform GO terms enrichment. As 'gene_info_file' we used https://ftp.ncbi.nih.gov/gene/DATA/GENE_INFO/Mammalia/Mus_musculus.gene_info.gz, as 'GO_obo_file' http://purl.obolibrary.org/obo/go/go-basic.obo, and as 'gene2GO_file' https://ftp.ncbi.nih.gov/gene/DATA/gene2go.gz (all downloaded on August 10th, 2022).

## Comparison between mouse and human programs

To compare mouse and human epithelial expression programs (*Pelka et al., 2021*), genes were mapped to mouse homologs using MGI homology information [subsection 'Mapping of mouse and human orthologs']. Mouse and human programs were then characterized by a single vector of mean expression per program in mouse gene space. Specifically, both mouse and human programs were defined such that their weighted sum approximates the expression profiles of the cells without any transformations. Programs and weights were normalized to sum to 1. To reduce batch effects (including species-specific ones), a background expression profile was defined for each species dataset as the pseudo-bulk epithelial expression profile in the respective scRNAs-seq data. Program and background profiles were normalized to 10,000 counts and the log ratio of the normalized program and background expression vectors was used to define a vector for each species. Pearson correlation coefficients were calculated for each pair of program vectors (mouse vs. human).

## Human expression program associations across mouse scRNA-seq

All sets of programs that were previously used to define human CRC tissue hubs (*Pelka et al., 2021*; epithelial, T/NK cells, and myeloid cells) were mapped to mouse genes with MGI homology information [subsection 'Mapping of mouse and human orthologs'] and then to mouse single cell data with TACCO, using TACCOs platform normalization to account for batch effects. The 'annotate' function in TACCO was used with OT as the core annotation method, on the comparable subsets of cells from mouse and human single-cell datasets (e.g. myeloid cells from mouse and human), with basic platform normalization, entropy regularization parameter epsilon 0.005, marginal relaxation parameter lambda of 0.1, and 4 iterations of bisectioning with a divisor of 3, and flat annotation prior distribution. The resulting probabilistic per-cell program annotations were aggregated to get probabilistic per-sample program annotations for all dysplastic mouse samples and CLR-transformed. For each pair of programs, the Pearson correlation coefficient was calculated on these transformed values.

## Annotating human scRNA-seq data with mouse-derived region information

Human scRNA-seq profiles (*Pelka et al., 2021*) were mapped to mouse gene space using MGI homology information [subsection 'Mapping of mouse and human orthologs']. Working in the same expression space, the 'annotate' function in TACCO with OT as core annotation method was used on the full human scRNA-seq and mouse Slide-seq dataset with basic platform normalization, entropy regularization parameter epsilon 0.005, marginal relaxation parameter lambda of 0.1, and 7 iterations of bisectioning with a divisor of 3, and 10-fold sub-clustering of the region annotations. The region transfer is done separately per compartment, with the Slide-seq compartment split as described above and the human scRNA-seq data split using the cell type annotation of the data. For validation, mapping was also performed with mouse scRNA-seq data, as well as mapping the region information from the mouse pucks back to themselves.

To test for enrichments of region annotations across disease state, region composition was aggregated to sample-level (for Slide-seq to four-way split pucks), CLR-transformed, and enrichment was calculated using a two-sided Welch's t-test. This was done for region annotation on human and mouse scRNA-seq data, and on the original and mapped region annotation on the mouse Slide-seq data.

## Cell-type associations across samples

To compare associations of cell types across samples in human and mouse scRNA-seq, the 'clMidwayPr' cell type annotation in the human data (*Pelka et al., 2021*) was aggregated to the same level as mouse cell type annotation, and then aggregated per sample and CLR-transformed. Pearson

correlation coefficients were calculated for every cell type pair for different subsets of samples: all samples, normal samples, dysplastic, and for human MMRd/MMRp samples.

## Epithelial program associations across human samples

To determine epithelial program associations across human samples, TACCO's 'annotate' function was used to annotate human epithelial scRNA-seq (after mapping to mouse orthologs using MGI homology information [subsection 'Mapping of mouse and human orthologs']) with mouse epithelial programs from mouse scRNA-seq data using OT as core method, basic platform normalization, entropy regularization parameter epsilon 0.005, marginal relaxation parameter lambda of 0.1, and 4 iterations of bisectioning with a divisor of 3. The remaining steps were performed as for cell-type association (subsection 'Cell-type associations across samples').

## Scoring epithelial mouse regions in mouse and human epithelial pseudo-bulk data

The published processed and filtered count matrices were used (where available) or instead raw count matrices for single cell/nucleus RNA seq data from *Pelka et al., 2021* (GEO accession number GSE178341; downloaded on August 8th, 2022), *Chen et al., 2021* (Synapse IDs syn27056096, syn27056097, syn27056098, syn27056099; downloaded on August 5th, 2022), *Khaliq et al., 2022* (GEO accession number GSE200997; downloaded on August 5th, 2022), *Becker et al., 2022* (GEO accession number GSE201348; downloaded on August 5th, 2022), *Zheng et al., 2022* (GEO accession number GSE161277; downloaded on August 5th, 2022; excluding 'blood' samples), *Che et al., 2021* (GEO accession number GSE178318; downloaded on August 5th, 2022; only 'CRC' and 'LM' samples) and *Joanito et al., 2022* (Synapse IDs syn26844072, syn26844073, syn26844078, syn26844087, syn26844111; downloaded on August 22nd, 2022; excluding the 'LymphNode' sample).

To subset the human single-cell data to epithelial cells, the epithelial annotation was used where readily available (*Pelka et al., 2021*; *Chen et al., 2021*). For the remaining datasets (*Becker et al., 2022*; *Che et al., 2021*; *Zheng et al., 2022*; *Khaliq et al., 2022*; *Joanito et al., 2022*), TACCOs tc.tl.annotate function was used with default parameters to transfer the 'cl295v11SubShort' annotation from *Pelka et al., 2021*, from which a compositional compartment annotation was constructed, and then a cell was assigned to the epithelial compartment if it had more than 95% epithelial fraction.

To correct for batch effects between the different data sources, first batches were defined by species times protokoll: 'mouse-10x3p', 'mouse-SlideSeq', 'human-10x3p' (*Pelka et al., 2021*; *Che et al., 2021*; *Zheng et al., 2022*; *Joanito et al., 2022*), 'human-10x5p' (*Khaliq et al., 2022*; *Joanito et al., 2022*), 'human-inDrop' (*Chen et al., 2021*), and 'human-snRNA' (*Becker et al., 2022*). Then TACCO's 'tc.pp.normalize_platform' function was used to determine per gene batch normalization factors using only the normal samples of one data source per batch (choosing the normal samples from Zheng for 'human-10x3p' and the normal 5' samples from Joanito for 'human-10x5p'). The resulting factors are then used to rescale the sample-by-gene count matrices for the full dataset per batch, that is including non-normal samples. The normalization factors are calculated with respect to an (arbitrarily chosen) normal reference dataset (*Zheng et al., 2022*).

The epithelial mouse region score was defined as the mean of the CLR-transformed expression values in the pseudo-bulk expression profile of the epithelial part of a dataset using the top 200 differentially expressed genes between all regions by a one-sided Fisher's exact test.

To account for species-specific biases (in-set vs. out-of-set prediction: the DEGs are calculated in mouse), the scores per region across samples were zero-centered and scaled to unit variance across all samples (including normal and non-normal samples and all batches) per species. A Principal Components Analysis (PCA) of the region scores across all species, batches, and samples was conducted and the values for the first PC were compared between different conditions using a two-sided Mann-Whitney U test with Benjamini-Hochberg FDR.

## Assessing the relationship between mouse regions and CMS tumor classification

We used the package 'CMSclassifier' (https://github.com/Sage-Bionetworks/CMSclassifier; *Bot, 2016*) referred to in *Guinney et al., 2015* to classify human pseudo-bulk CRC profiles from all samples which were not normal or unaffected from the human studies above into CMS classes. We determine

the enrichment (Benjamini-Hochberg FDR, two-sided Welch's t test) of the same mouse region scores in the CMS classes.

## Assessing the relation between mouse regions and clinical endpoints in human bulk RNA-seq

Published RNA-seq data from the COAD and READ cohorts of TCGA PanCancerAtlas (*Liu et al., 2018*) were used. Mouse region scores were defined as the mean of the log1p-transformed, zero-centered, and scaled expression values in the bulk expression profile using the top 200 differentially expressed genes between the malignant mouse regions (6, 8, and 11) by a one-sided Fisher's Exact test (comparing each of the three regions to the other two). Scores were stratified into quartiles. PFI and OS were compared between patients with tumors whose scores were in the lowest and highest quartiles using the Logrank test as implemented in the lifelines package (*Davidson-Pilon, 2019*), followed by Benjamini-Hochberg FDR.

## Compositional enrichment analyses

Enrichments on compositional data (cell type compositions, etc.) were evaluated with a two-sided Welch's t test on sample level using CLR-transformed compositions followed by Benjamini-Hochberg FDR. For the enrichment of tdTomato, counts and ALR-transformation were used instead with all non-tdTomato counts used as reference compartment. Enrichment analyses were performed using TACCO's 'enrichments' function.

## Code availability

The analysis code is available on GitHub (https://github.com/simonwm/mouseCRC copy archived at *Mages, 2025*).

## Acknowledgements

We thank N Friedman, I Benhar, N Habib, M Biton, K Geiger-Schuller, JC Hütter, B Dumitrascu, E Baker and A Greenwald for helpful discussions. We thank C McCabe, O Kuksenko and I Barrera for technical assistance. We thank P Yadollahpour, E Dhaval, G Smith-Rosario, S Vickovic, D Schapiro, S Farhi, D Abbondanza, A Segerstolpe and T Biancalani for their important contribution to this project. We thank L Gaffney and A Hupalowska for help with figure preparation. SM was supported by a DFG research fellowship (MA 9108/1–1), JK was supported by a HFSP long term fellowship (LT000452/2019 L), AR was a Howard Hughes Medical Institute (HHMI) Investigator when conducting this work. Work was supported by the Klarman Cell Observatory, a CEGS grant (5RM1HG006193-09) from the NHGRI, the NIH/NIAID (grants 1U24 CA180922, 1U19 MH114821, 1RC2 DK114784), the MIT Ludwig Center, the Manton Family Foundation, and HHMI (AR); Azrieli Foundation Early Career Faculty Fellowship, and an ISF Research Grant (1079/21) (MN), the Center for Interdisciplinary Data Science Research at the Hebrew University of Jerusalem (NM and MN), the Israeli Council for Higher Education Ph.D. fellowship (NM), SU2C Peggy Prescott Early Career Scientist Award PA-6146, SU2C Phillip A Sharp Award SU2C-AACR-PS-32 and NIH/NCI R00CA259511 (KP), NIH/NCI R01 CA208756; Arthur, Sandra, and Sarah Irving Fund for Gastrointestinal Immuno-Oncology (NH), NIH/NCI R01CA257523, MIT Stem Cell Initiative (Foundation MIT) (OY), and NIH R37CA259363, R21CA256414, R21DK125911, R41EB032693, R01CA254108, R01CA256530, and R01CA244359; DOD W81XWH-20-1-0203; and a Duke-NC State Translational Research Grant (JR). The authors gratefully acknowledge LMU Klinikum for providing computing resources on their Clinical Open Research Engine (CORE) and the Bioinformatic Core Facility of the Biomedical Center Munich for providing computing resources on their HPC system.

## Additional information

### Competing interests

Karin Pelka: K.P. reports consulting fees from Santa Ana Bio, Inc and GV Management Company, L.L.C. Arnav Mehta: A.M. has served a consultant/advisory role for Third Rock Ventures, Asher

Biotherapeutics, AbataTherapeutics, ManaT Bio, Flare Therapeutics, venBio Partners, BioNTech, Rheos Medicines andCheckmate Pharmaceuticals, is currently a part-time Entrepreneur in Residence at Third RockVentures, is an equity holder in ManaT Bio, Asher Biotherapeutics and Abata Therapeutics, andhas received research funding support from Bristol-Myers Squibb. Genevieve M Boland: G.M.B. has sponsored researchagreements with InterVenn Biosciences, Palleon Pharmaceuticals, Olink Proteomics, and TeikoBio. G.M.B. is a consultant for Ankyra Therapeutics and InterVenn Bio. G.M.B. has been onscientific advisory boards for Merck, Iovance, Nektar Therapeutics, Instil Bio, and AnkyraTherapeutics. G.M.B. holds equity in Ankyra Therapeutics. J.L. Morgane Rouault: M.R. hold equity in10xGenomics. Hacohen Nir: N.H. holds equity in BioNTech and is a founder of Related Sciences/DangerBio. Fei Chen: F.C. is a founder and holds equity in Curio Biosciences. Omer Yilmaz: O.Y. holds equity and is a SAB member1232of AVA Lifesciences. Orit Rozenblatt-Rosen: O.R.-R. are co-inventors on patent applications filed by the Broad Institute for inventions related to single cell genomics. O.R.-R. has given numerous lectures on thesubject the subject of single cell genomics to a wide variety of audiences and in some cases, has receivedremuneration to cover time and costs. O.R.-R. is an employee of Genentech since October 19,and has equity in Roche. Aviv Regev: A.R. is a co-inventors on patent applications filed by the Broad Institute for inventions related to single cell genomics. A.R. is a co-founder and equity holder of Celsius Therapeutics, anequity holder in Immunitas, and was an SAB member of ThermoFisher Scientific, SyrosPharmaceuticals, Neogene Therapeutics and Asimov until July 31, 2020. From August 1, 2020,A.R. is an employee of Genentech and has equity in Roche. The other authors declare that no competing interests exist.

## Funding

| Funder | Grant reference number | Author |
|--------|------------------------|--------|
| Human Frontier Science Program | LT000452/2019-L | Johanna Klughammer |
| Howard Hughes Medical Institute | | Aviv Regev |
| Klarman Cell Observatory, Broad Institute | | Aviv Regev |
| National Human Genome Research Institute | 5RM1HG006193-09 | Aviv Regev |
| National Institute of Allergy and Infectious Diseases | 1U24 CA180922 | Aviv Regev |
| Ludwig Center for Molecular Oncology | | Aviv Regev |
| Manton Foundation | | Aviv Regev |
| Azrieli Foundation | | Mor Nitzan |
| Israel Science Foundation | 1079/21 | Mor Nitzan |
| National Cancer Institute | NIH/NCI R00CA259511 | Karin Pelka Mor Nitzan |
| National Cancer Institute | NIH/NCI R01 CA208756 | Hacohen Nir |
| National Cancer Institute | NIH/NCI R01CA257523 | Omer Yilmaz |
| National Institutes of Health | NIH R37CA259363 | Jatin Roper |
| Deutsche Forschungsgemeinschaft | research fellowship MA 9108/1-1 | Simon Mages |
| Center for Interdisciplinary Data Science Research at the Hebrew University of Jerusalem | | Noa Moriel Mor Nitzan |
| The Israeli Council for Higher Education Ph.D fellowship | | Noa Moriel |

| Funder | Grant reference number | Author |
| --- | --- | --- |
| SU2C Peggy Prescott Early Career Scientist Award | PA-6146 | Karin Pelka |
| SU2C Phillip A. Sharp Award | SU2C-AACR-PS-32 | Karin Pelka |
| Arthur, Sandra, and Sarah Irving Fund for Gastrointestinal Immuno-Oncology | | Omer Yilmaz |
| MIT Stem Cell Initiative | | Omer Yilmaz |
| DOD Prostate Cancer Research Program | W81XWH-20-1-0203 | Jatin Roper |
| Duke-NC State Translational Research Grant | | Jatin Roper |
| National Institute of Allergy and Infectious Diseases | 1U19 MH114821 | Aviv Regev |
| National Institute of Allergy and Infectious Diseases | 1RC2 DK114784 | Aviv Regev |
| National Institutes of Health | R21CA256414 | Jatin Roper |
| National Institutes of Health | R21DK125911 | Jatin Roper |
| National Institutes of Health | R41EB032693 | Jatin Roper |
| National Institutes of Health | R01CA254108 | Jatin Roper |
| National Institutes of Health | R01CA256530 | Jatin Roper |
| National Institutes of Health | R01CA244359 | Jatin Roper |

The funders had no role in study design, data collection and interpretation, or the decision to submit the work for publication.

## Author contributions

Inbal Avraham-Davidi, Conceptualization, Formal analysis, Investigation, Visualization, Methodology, Writing – original draft, Project administration, Writing – review and editing, Corresponding author; Simon Mages, Conceptualization, Software, Formal analysis, Investigation, Visualization, Methodology, Writing – original draft, Project administration, Writing – review and editing, Equal contribution with Inbal Avraham-Davidi and Johanna Klughammer; Johanna Klughammer, Conceptualization, Software, Formal analysis, Visualization, Methodology, Writing – original draft, Writing – review and editing, Equal contribution with Inbal Avraham-Davidi and Simon Mages; Noa Moriel, Conceptualization, Software, Formal analysis, Investigation, Methodology, Writing – review and editing; Shinya Imada, Conceptualization, Validation, Investigation, Methodology, Writing – review and editing; Matan Hofree, Software, Formal analysis, Methodology, Writing – review and editing; Evan Murray, Investigation, Methodology; Jonathan Chen, Genevieve M Boland, Validation, Investigation, Methodology, Writing – review and editing; Karin Pelka, Resources, Validation, Investigation, Methodology; Arnav Mehta, Formal analysis, Validation, Investigation, Methodology, Writing – review and editing; Toni Delorey, Leah Caplan, Investigation, Methodology, Writing – review and editing; Danielle Dionne, Methodology, Writing – review and editing; Robert Strasser, Jana Lalakova, Anezka Niesnerova, Hao Xu, Methodology; Morgane Rouault, Methodology, Project administration; Itay Tirosh, Resources, Project administration, Writing – review and editing; Hacohen Nir, Resources, Supervision, Writing – review and editing; Fei Chen, Resources, Supervision, Methodology, Writing – review and editing; Omer Yilmaz, Resources, Supervision, Investigation, Methodology, Writing – original draft, Project

administration, Writing – review and editing; Jatin Roper, Resources, Formal analysis, Validation, Investigation, Methodology, Writing – original draft, Writing – review and editing; Orit Rozenblatt-Rosen, Resources, Formal analysis, Supervision, Funding acquisition, Investigation, Methodology, Writing – original draft, Project administration, Writing – review and editing; Mor Nitzan, Conceptualization, Resources, Software, Formal analysis, Supervision, Investigation, Methodology, Writing – original draft, Writing – review and editing; Aviv Regev, Conceptualization, Resources, Formal analysis, Supervision, Funding acquisition, Investigation, Methodology, Writing – original draft, Project administration, Writing – review and editing

### Author ORCIDs
Inbal Avraham-Davidi ⓘ https://orcid.org/0000-0001-7118-9179
Simon Mages ⓘ https://orcid.org/0000-0003-1447-6811
Johanna Klughammer ⓘ http://orcid.org/0000-0002-3628-9278
Toni Delorey ⓘ https://orcid.org/0000-0001-6614-3803
Leah Caplan ⓘ https://orcid.org/0000-0002-6594-1608
Itay Tirosh ⓘ https://orcid.org/0000-0001-5477-2987
Omer Yilmaz ⓘ https://orcid.org/0000-0002-7577-4612
Aviv Regev ⓘ https://orcid.org/0000-0003-3293-3158

### Ethics
Mice were housed in the animal facility at the Koch Institute for Integrative Cancer Research at MIT. All animal studies described in this study were approved by the MIT Institutional Animal Care and Use Committee (Protocol 1213-106-16).

Reviewer #2 (Public review): https://doi.org/10.7554/eLife.104815.3.sa1
Author response https://doi.org/10.7554/eLife.104815.3.sa2

---

## Additional files

### Supplementary files
MDAR checklist

Supplementary file 1. Differentially expressed genes and pathway enrichment in stromal cells.

Supplementary file 2. Epithelial program expression in malignant like regions.

Supplementary file 3. Probe list for Multiplex in situ RNA Analysis.

### Data availability
All data generated in this project is deposited on the Single Cell Portal and is available under the accession SCP1891 (https://singlecell.broadinstitute.org/single_cell/study/SCP1891). The raw scRNA-seq and Slide-seq data is also deposited on GEO under the GSE260801(https://www.ncbi.nlm.nih.gov/geo/query/acc.cgi?acc=GSE260801). The public human scRNA-seq datasets which were reused in this project can be found on GEO under accession numbers GSE178341, GSE200997,GSE201348, GSE161277, GSE178318 and on Synapse under the IDs syn27056096, syn27056097, syn27056098, syn27056099, syn26844072, syn26844073, syn26844078, syn26844087, syn26844111.

The following dataset was generated:

| Author(s) | Year | Dataset title | Dataset URL | Database and Identifier |
| --- | --- | --- | --- | --- |
| Mages S | 2024 | Spatially defined multicellular functional units in colorectal cancer revealed from single cell and spatial transcriptomics | https://www.ncbi.nlm.nih.gov/geo/query/acc.cgi?acc=GSE260801 | NCBI Gene Expression Omnibus, GSE260801 |

The following previously published datasets were used:

| Author(s) | Year | Dataset title | Dataset URL | Database and Identifier |
| --- | --- | --- | --- | --- |
| Li J, Che L | 2021 | A Single-Cell Atlas of Liver Metastases of Colorectal Cancer Reveals the Reprogramming of the Tumor Microenvironment in Response to Preoperative Chemotherapy | https://www.ncbi.nlm.nih.gov/geo/query/acc.cgi?acc=GSE178318 | NCBI Gene Expression Omnibus, GSE178318 |
| Zheng X, Song J, Liu X, Yu C, Zhou Z, Shi H | 2021 | Dissecting the Epithelial and Immune Evolution during Colorectal Carcinogenesis by Single-Cell RNA Sequencing | https://www.ncbi.nlm.nih.gov/geo/query/acc.cgi?acc=GSE161277 | NCBI Gene Expression Omnibus, GSE161277 |
| Becker WR, Nevins SA, Chen DC, Chiu R, Horning A, Guha TK, Laquindanum R, Mills M, Chaib H, Ladabaum U, Longacre T, Shen J, Esplin ED, Kundaje A, Ford JM, Curtis C, Snyder MP, Greenleaf WJ | 2022 | Single-cell analyses define a continuum of cell state and composition changes in the malignant transformation of polyps to colorectal cancer [scRNA-seq] | https://www.ncbi.nlm.nih.gov/geo/query/acc.cgi?acc=GSE201348 | NCBI Gene Expression Omnibus, GSE201348 |
| Khaliq AM, Masood A | 2022 | Refining Colorectal Cancer Classification and Clinical Stratification Through a Single-Cell Atlas | https://www.ncbi.nlm.nih.gov/geo/query/acc.cgi?acc=GSE200997 | NCBI Gene Expression Omnibus, GSE200997 |
| Pelka K, Chen JH, Anderson AC, Rozenblatt-Rosen O, Regev A, Hachoen N | 2021 | A Single Cell Atlas of MMRd and MMRp Colorectal Cancer | https://www.ncbi.nlm.nih.gov/geo/query/acc.cgi?acc=GSE178341 | NCBI Gene Expression Omnibus, GSE178341 |

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
