## [Editor Report · eLife Assessment]

This work presents a **valuable** resource combining scRNA-seq and spatial transcriptomics studies to map mouse pre-clinical models of colorectal cancer, identifying distinct cellular programs and microenvironments that could enhance patient stratification and therapeutic approaches in colorectal cancer. While the evidence provided in the manuscript are not fully validated, these **solid** data were collected and analyzed using a validated methodology that will be of interest to the community in future studies.

---

## [Referee Report · Reviewer #2 (Public review)]

In their study, Avraham-Davidi et al. combined scRNA-seq and spatial mapping studies to profile two preclinical mouse models of colorectal cancer: Apcfl/fl VilincreERT2 (AV) and Apcfl/fl LSL-KrasG12D Trp53fl/fl Rosa26LSL-tdTomato/+ VillinCreERT2 (AKPV). In the first part of the manuscript, the authors describe the analysis of the normal colon and dysplastic lesions induced in these models following tamoxifen injection. They highlight broad variations in immune and stromal cell composition within dysplastic lesions, emphasizing the infiltration of monocytes and granulocytes, the accumulation of IL-17+gdT cells and the presence of a distinct group of endothelial cells. A major focus the study is the remodeling of the epithelial compartment, where most significant changes are observed. Using no-negative matrix factorization, the authors identify molecular programs of epithelial cell functions, emphasizing stemness, Wnt signaling, angiogenesis and inflammation as majors features associated with dysplastic cells. They conclude that findings from scRNA-seq analyses in mouse models are transposable to human CRC. In the second part of the manuscript, the authors aim to provide the spatial contexture for their scRNA-seq findings using Slide-seq and TACCO. They demonstrate that dysplastic lesions are disorganized and contain tumor-specific regions, which contextualize the spatial proximity between specific cell states and gene programs. Finally, they claim that these spatial organizations are conserved in human tumors and associate region-based gene signatures with patient outcome in public datasets. Overall, the data were collected and analyzed using solid and validated methodology to offer a useful resource to the community.

Main comments:

(1) Clarity. The manuscript would benefit from a substantial reorganization to improve clarity and accessibility for a broad readership. The text could be shortened and the number of figure panels reduced to emphasize the novel contributions of this work while minimizing extensive discussions on general and expected findings, such as tissue disorganization in dysplastic lesions. Additionally, figure panels are not consistently introduced in the correct order, and some are not discussed at all (e.g., Fig. S1D; Fig. 3C is introduced before Fig. 3A; several panels in Fig. 4 are not discussed). The annotation of scRNA-seq cell states is insufficiently explained, with no corresponding information about associated genes provided in the figures or tables. Multiple annotations are used to describe cell groups (e.g., TKN01 = γδ T and CD8 T, TKN05 = γδT_IL17+), but these are not jointly accessible in the figures, making the manuscript challenging to follow. It is also not clear what is the respective value of the two mouse models and timepoints of tissue collection in the analysis.

(2) Novelty. While the study is of interest, it does not present major findings that significantly advance the field or motivate new directions and hypotheses. Many conclusions related to tissue composition and patient outcomes, such as the epithelial programs of Wnt signaling, angiogenesis, and stem cells, are well-established and not particularly novel. Greater exploration of the scRNA-seq data beyond cell type composition could enhance the novelty of the findings. For instance, several tumor microenvironment clusters uniquely detected in dysplastic lesions (e.g., Mono2, Mono3, Gran01, Gran02) are identified, but no further investigation is conducted to understand their biological programs, such as applying nNMF as was done for epithelial cells. Additional efforts to explore precise tissue localization and cellular interactions within tissue niches would provide deeper insights and go beyond the limited analyses currently displayed in the manuscript.

(3) Validation. Several statements made by the authors are insufficiently supported by the data presented in the manuscript and should be nuanced in the absence of proper validation. For example: (1.) RNA velocity analyses: The conclusions drawn from these analyses are speculative and need further support. (2.) Annotations of epithelial clusters as dysplastic: These annotations could have been validated through morphological analyses and staining on FFPE slides. (3.) Conservation of mouse epithelial programs in human tumors: The data in Figure S5B does not convincingly demonstrate enrichment of stem cell program 16 in human samples. This should be more explicitly stated in the text, given the emphasis placed on this program by the authors. (4.) Figure S6E: Cluster Epi06 is significantly overrepresented in spatial data compared to scRNA-seq, yet the authors claim that cell type composition is largely recapitulated without further discussion, which reduces confidence in other conclusions drawn.

Furthermore, stronger validation of key dysplastic regions (regions 6, 8, and 11) in mouse and human tissues using antibody-based imaging with markers identified in the analyses would have considerably strengthened the study. Such validation would better contextualize the distribution, composition, and relative abundance of these regions within human tumors, increasing the significance of the findings and aiding the generation of new pathophysiological hypotheses.

Comments on revisions:

The authors have improved the clarity of the manuscript and responded adequately to all my initial comments.

I don't have any other comments. Congratulations to the authors on this work.

---

## [Author Response]

The following is the authors’ response to the original reviews.

**Reviewer #1 (Public review):**
Summary:The authors conducted a spatial analysis of dysplastic colon tissue using the Slide-seq method. Their main objective is to build a detailed spatial atlas that identifies distinct cellular programs and microenvironments within dysplastic lesions. Next, they correlated this observation with clinical outcomes in human colorectal cancer.Strengths:The work is a good example of utilising spatial methods to study different tumour models. The authors identified a unique stem cell program to understand tumours gently and improve patient stratification strategies.Weaknesses:However, the study's predominantly descriptive nature is a significant limitation. Although the spatial maps and correlations between cell states are interesting observations, the lack of functional validation-primarily through experiments in mouse models-weakens the causal inferences regarding the roles these cellular programs play in tumour progression and therapy resistance.

We thank the reviewer for this comment. Indeed, functional validation to pin down causal dependencies and a more thorough investigation of tumor progression and therapy resistance both in mouse model as well as human patients and/or patient derived samples would broaden the insights to be gained from this work. Unfortunately, this is beyond the scope of this study.

The authors also missed an opportunity to link the mutational status of malignant cells with the cellular neighbourhoods.

The data reported in this study only contains spatial data for one mouse model (AV). As spatial data for the other model (AKPV) is missing, it is not possible to link the mutational type of the model with the cellular neighborhoods. We did investigate whether there is extra somatic mutational heterogeneity in the AV data, both regarding single nucleotide variations (SNVs) and copy number variations (CNVs). But at the time when the mice were sacrificed (after 3 weeks) there was no significant mutational heterogeneity discoverable.

Overall, the study contributes to profiling the dysplastic colon landscape. The methodologies and data will benefit the research community, but further functional validation is crucial to validate the biological and clinical implications of the described cellular interactions.
**Reviewer #2 (Public review):**
In their study, Avraham-Davidi et al. combined scRNA-seq and spatial mapping studies to profile two preclinical mouse models of colorectal cancer: Apcfl/fl VilincreERT2 (AV) and Apcfl/fl LSL-KrasG12D Trp53fl/fl Rosa26LSL-tdTomato/+ VillinCreERT2 (AKPV). In the first part of the manuscript, the authors describe the analysis of the normal colon and dysplastic lesions induced in these models following tamoxifen injection. They highlight broad variations in immune and stromal cell composition within dysplastic lesions, emphasizing the infiltration of monocytes and granulocytes, the accumulation of IL-17+gdT cells, and the presence of a distinct group of endothelial cells. A major focus of the study is the remodeling of the epithelial compartment, where the most significant changes are observed. Using non-negative matrix factorization, the authors identify molecular programs of epithelial cell functions, emphasizing stemness, Wnt signaling, angiogenesis, and inflammation as major features associated with dysplastic cells. They conclude that findings from scRNA-seq analyses in mouse models are transposable to human CRC. In the second part of the manuscript, the authors aim to provide the spatial context for their scRNA-seq findings using Slide-seq and TACCO. They demonstrate that dysplastic lesions are disorganized and contain tumor-specific regions, which contextualize the spatial proximity between specific cell states and gene programs. Finally, they claim that these spatial organizations are conserved in human tumors and associate region-based gene signatures with patient outcomes in public datasets. Overall, the data were collected and analyzed using solid and validated methodology to offer a useful resource to the community.Main comments:(1) ClarityThe manuscript would benefit from a substantial reorganization to improve clarity and accessibility for a broad readership. The text could be shortened and the number of figure panels reduced to emphasize the novel contributions of this work while minimizing extensive discussions on general and expected findings, such as tissue disorganization in dysplastic lesions. Additionally, figure panels are not consistently introduced in the correct order, and some are not discussed at all (e.g., Figure S1D; Figure 3C is introduced before Figure 3A; several panels in Figure 4 are not discussed). The annotation of scRNA-seq cell states is insufficiently explained, with no corresponding information about associated genes provided in the figures or tables. Multiple annotations are used to describe cell groups (e.g., TKN01 = γδ T and CD8 T, TKN05 = γδT_IL17+), but these are not jointly accessible in the figures, making the manuscript challenging to follow. It is also not clear what is the respective value of the two mouse models and time points of tissue collection in the analysis.

We thank the reviewer for this suggestion. We clarified and simplified the revised manuscript, however we believe that the current discussions are an important part of the manuscript and would be useful to readers. We reordered panels in Figures S1 and 3 to align with their appearance in the manuscript. We kept the order of other panels as it is to keep both context and coherence of those figures intact. We changed the way we reference cell clusters in the manuscript to better align with the naming scheme introduced in Figure 1B. The respective value of the two mouse models as well as the time points of tissue collection are described in lines 108-120 of the manuscript.

(2) NoveltyWhile the study is of interest, it does not present major findings that significantly advance the field or motivate new directions and hypotheses. Many conclusions related to tissue composition and patient outcomes, such as the epithelial programs of Wnt signaling, angiogenesis, and stem cells, are well-established and not particularly novel. Greater exploration of the scRNA-seq data beyond cell type composition could enhance the novelty of the findings. For instance, several tumor microenvironment clusters uniquely detected in dysplastic lesions (e.g., Mono2, Mono3, Gran01, Gran02) are identified, but no further investigation is conducted to understand their biological programs, such as applying nNMF as was done for epithelial cells. Additional efforts to explore precise tissue localization and cellular interactions within tissue niches would provide deeper insights and go beyond the limited analyses currently displayed in the manuscript.

We thank the reviewer for this comment. Our study aimed to spatially characterize the tumor microenvironment, with scRNA-seq analysis serving to support this spatial characterization.

Due to technical limitations—such as the number of samples and the limited capture efficiency of Slide-seq—the resolution of immune cell identification in our spatial analysis is constrained. Additionally, while immune and stromal cells formed distinct clusters, epithelial cells exhibited a continuum that was better captured using nNMF.

Lastly, our manuscript provides a general characterization of monocyte and granulocyte populations in scRNA-seq (line 144) and their spatial microenvironments (line 400). We believe that additional analyses of these populations would be beyond the scope of this study and could place an unnecessary burden on the reader. Instead, we suggest that such analyses be explored in future studies.

We remark that we analyzed tissue localization for two entirely different spatial transcriptomics assays (Slide-seq and Cartana) at the resolution of cell types and programs, which was feasible within the constraints of the sparsity, gene panel and sample size in the experiments. A future potential path to further increase the resolution of investigation in this dataset is to include other datasets, e.g. by the emerging transformer-based spatial transcriptomics integration methods.

We also remark that the manuscript already includes an investigation of cellular interactions within tissue niches based on COMMOT (Fig 4k, Fig S8i, Supp Item 4).

(3) ValidationSeveral statements made by the authors are insufficiently supported by the data presented in the manuscript and should be nuanced in the absence of proper validation. For example:(a) RNA velocity analyses: The conclusions drawn from these analyses are speculative and need further support.

We thank the reviewer for this comment. We clarified that our conclusions from the RNA velocity analysis need further support by experimental validation (lines 223-225), which is outside the scope of the current study.

(b) Annotations of epithelial clusters as dysplastic: These annotations could have been validated through morphological analyses and staining on FFPE slides.

We thank the reviewer for this comment. While this could have been a possible approach, our study primarily relies on scRNA-seq, which does not preserve tissue morphology, and Slide-seq of fresh tissue, where such an analysis is particularly challenging.

(c) Conservation of mouse epithelial programs in human tumors: The data in Figure S5B does not convincingly demonstrate the enrichment of stem cell program 16 in human samples. This should be more explicitly stated in the text, given the emphasis placed on this program by the authors.

We thank the reviewer for pointing this out. We clarified the section about the stem cell program 16 and references to Figures S5A and S5B (lines 269-274): while we do see correlation in the definition of human programs with the mouse stem cell program (Figure S5A), we do not see a correlated expression of the stem cell program across human and mouse (Figure S5B).

(d) Figure S6E: Cluster Epi06 is significantly overrepresented in spatial data compared to scRNA-seq, yet the authors claim that cell type composition is largely recapitulated without further discussion, which reduces confidence in other conclusions drawn.

We thank the reviewer for this remark. Indeed, Epi06 was a cluster which drew our attention during early analyses for its mixed expression profiles with contributions of vastly different cell types. We concluded that this is best explained by doublets, but we cannot rule out (partial) non-doublet explanations (e.g. undifferentiated cells). As doublet detection with Scrublet did not flag those cells as doublets, we kept these cells in the workflow, but excluded them from further interpretation. While in the previous version of the manuscript we only shortly hinted to this in figure legend 2A ("Cluster Epi06: doublets (not called by Scrublet)"), we expanded on this in the methods section of the revised manuscript (lines 863-869). Given the doublet interpretation, the observation that this cluster is significantly overrepresented in the annotation of the spatial data is not surprising as this annotation comes from the decomposition of compositional data which contains contributions of multiple cells per Slide-seq bead which are structurally very similar to doublets. While Epi06 appears enriched in S6E when comparing Slide-Seq to scRNA-seq, there are multiple technical cross platform differences, including different per-gene sensitivities or capture biases for certain cell types (e.g. stromal cells suffering more from dissociation in scRNA compared to Slide-Seq). We believe that comparisons between disease states within a single platform are more biologically meaningful, like the comparison between normal and premalignant tissue, which is presented in Figure S6G. To increase confidence in the analysis and to assess whether intra-platform biological conclusions are affected by the inclusion/exclusion of Epi06, we recreated Figure S6G for a Slide-Seq cell type annotation without Epi06 in the reference (see Author response image 1). Even though Epi06 is missing in that annotation, the strong enrichments are consistently preserved between the two analysis variants, while as expected some less significant enrichments with larger FDR values are not preserved.

**Author response image 1. sa2fig1:** Significance (FDR, color bar, two-sided Welch’s t test on CLR-transformed compositions) of enrichment (red) or depletion (blue) of cell clusters (rows) in normal (N) or AV (AV) tissues based on Slide-seq (“spatial”) data or scRNA-seq ("sc”) including (A) or excluding (B) Epi06 in the reference for annotating the Slide-Seq data (A is identical to Figure S6G in the manuscript).

Furthermore, stronger validation of key dysplastic regions (regions 6, 8, and 11) in mouse and human tissues using antibody-based imaging with markers identified in the analyses would have considerably strengthened the study. Such validation would better contextualize the distribution, composition, and relative abundance of these regions within human tumors, increasing the significance of the findings and aiding the generation of new pathophysiological hypotheses.

We agree with the reviewer with their assessment that validation by antibody-based imaging (or other spatial proteomics data) would have been useful follow-up experiments, yet these are beyond the scope of the current study.
**Reviewer #1 (Recommendations for the authors):**
AV and AKPV have different oncogenic mutations, and their impact on spatial neighbourhoods is unclear. Can authors perform an analysis to understand the contribution of oncogenic mutations on the spatial landscape of CRC?

The data reported in this study only contains spatial data for one mouse model (AV). As spatial data for the other model (AKPV) is missing, it is not possible to comparatively link the mutational type of the model with the spatial landscape.